# Application of enhanced benders decomposition algorithm in circular assembly line balancing problem with task splitting

Panfei Li[1]*, Chongxing Ji[1,2]

**1** School of Artificial Intelligence, Dongguan City University, Dongguan, Guangdong, China, **2** Faculty of Applied Sciences, Macao Polytechnic University, Macao SAR, China

* lipanfei@dgcu.edu.cn

## Abstract

The advent of the assembly line marked a significant technological innovation in the manufacturing industry, substantially enhancing production efficiency. Today, this production system is extensively adopted by numerous manufacturing enterprises. This paper introduces the Circular Assembly Line Balancing Problem with Task-Splitting (CALBP-TS), a novel NP-hard optimization challenge characterized by closed-loop topology, station revisitation, fixed-position machines, and collaborative task execution. To address its high-dimensional complexity, we propose an Enhanced Benders Decomposition (EBD) framework that decomposes CALBP-TS into a workload-balancing master problem (MP) addressing worker-process assignment and task-splitting using a rigorous linearization theorem and a feasibility-checking subproblem (SP) handling spatio-temporal constraints via dummy process encoding. Key algorithmic accelerators comprise a Heuristic Infeasibility Proof (HIP) for rapid solution screening and Enhanced Benders Cuts (EBC) derived from infeasibility analysis, both integrated with integrated with Local Branching. Validated on 60 real-world instances from Huawei, EBD achieves average runtime reductions of 97.8%, 69.2%, and 48.4% compared to MILP, GA+LP, and Greedy+LP baselines, respectively, while improving solution quality by up to 41.3%. Ablation studies confirm that HIP and EBC collectively enhance computational efficiency by 13.7%. Our methodology facilitates optimal resource utilization in space-constrained circular production systems.

## 1 Introduction

An assembly line represents a fundamental manufacturing approach where products are assembled sequentially at workstations, with workers assigned to perform specific operations or tasks [1]. In recent years, a considerable body of research on the Assembly Line Balancing Problem (ALBP) has predominantly focused on traditional straight-line configurations [2–4]. Although straight-line configurations offer

**Data availability statement:** Data is available from the following link (https://github.com/LiPanfei-Lab/CALBP-TS).

**Funding:** The author(s) received no specific funding for this work.

**Competing interests:** Industrial System Optimization, Operations Research, Deep Learning, and Intelligent Control.

advantages in structural simplicity and mathematical tractability, they inherently impede worker collaboration and auxiliary equipment circulation—factors increasingly essential in modern manufacturing systems. As noted in reference [5], U-shaped layout is intended to increase flexibility and productivity, the stations are not arranged in a straight line but in a U-form such that workers can perform tasks on two stations within the same cycle in so-called cross-over workplaces. Consequently, substantial research has recently emerged on U-shaped assembly line balancing problems, exemplified by the methodologies proposed in references [6–11].

In this paper, we propose and study the Circular Assembly Line Balancing Problem while considering Task-Splitting (CALBP-TS). This problem constitutes a novel variant and extension of the classical U-shaped Assembly Line Balancing Problem (ULBP). Whereas the classical ULBP permits workload sharing only among stations within the same U-shaped line (typically across different sides), our model extends this capability to enable workers to share workload across distinct stations belonging to different U-shaped lines [5]. The proposed Circular assembly lines (CALs) represent an advanced manufacturing configuration characterized by a closed-loop topology where stations are arranged in a circular layout, enabling processes to revisit stations across multiple production cycles. This structure is adopted in space-constrained industries (e.g., automotive, electronics) owing to its superior footprint efficiency and ability to accommodate fixed-position machines– immovable resources imposing constraints on conventional layouts. Unlike linear or U-shaped lines, Circular Assembly Lines (CALs) enable non-linear task sequencing by leveraging their closed-loop topology. This circumferencial movement allows workpieces to revisit stations or bypass others en route to fixed-machine locations, effectively decoupling task execution order from physical station sequence. This flexibility is further enhanced by proximity-based adjacency (including diametrically opposite stations), making CALs particularly suited for complex, hierarchical assemblies. However, these advantages introduce NP-hard challenges distinct from conventional ALBPs: 1) Revisitation overhead: Circumferential travel increases transport time, necessitating strict cycle limits (enforced by $max\_cycle\_count$); 2) Spatial dynamics: The circular layout complicates synchronization as workpieces dynamically traverse the loop, potentially skipping stations to reach required machines; 3) Task splitting (a core feature of CALBP-TS): Distributing bottleneck processes among multiple workers (up to $max\_worker\_per\_oper$) improves balance but significantly amplifies combinatorial complexity, task synchronization overhead, and resource conflicts; 4) Worker mobility constraints: Assigning non-adjacent stations (limited by $max\_station\_per\_worker$) incurs movement penalties not present in unidirectional systems. Collectively, the inherent cyclicity, dynamic routing, task splitting, and worker mobility constraints exponentially increase solution-space dimensionality, demanding novel optimization approaches.

The herein proposed CALBP-TS is a NP-hard combinatorial optimization problem, its primary objective is to determine the assignment scheme of processes, workers, stations, and machines under process rules, worker capabilities, station constraints, machine specifications, and other operational restrictions, while minimizing the maximum worker workload and workload volatility rate. In Fig 1, we schematically

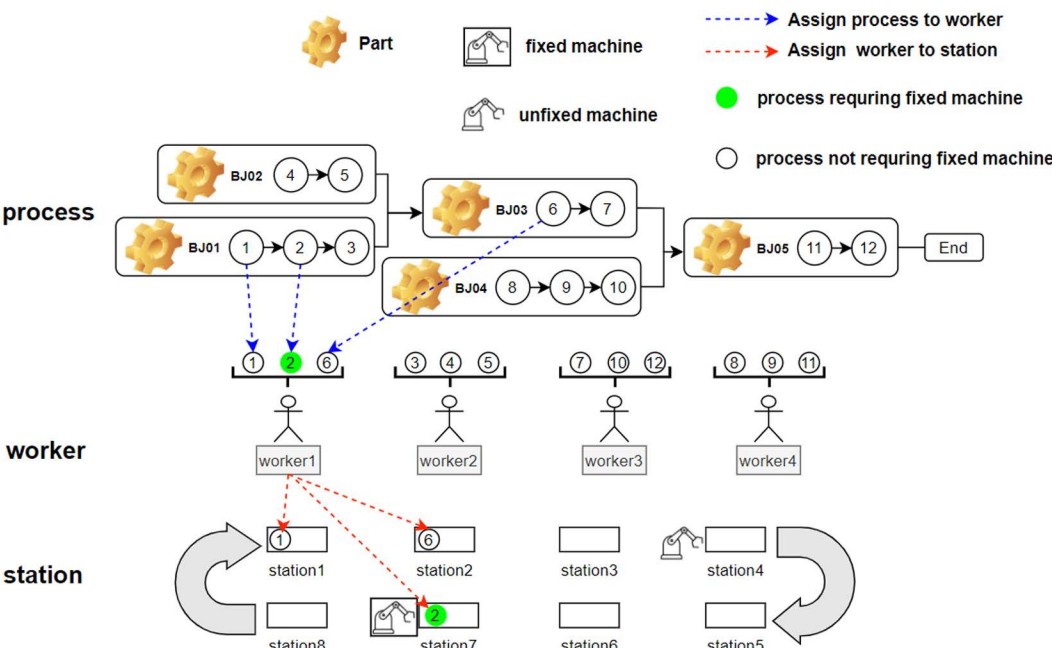

**Fig 1. Layout example of circular assembly line balancing problem.**

represent the main decisions involved in the CALBP-TS. A directed acyclic graph (DAG) models a task set (node from 1 to 12) and its precedence constraints. The tasks are partitioned into disjoint subsets, with each subset assigned to a worker (denoted by blue dashed directed edges). Subsequently, workers are assigned to stations (denoted by red dashed directed edges). Both task partitioning and worker-station assignment decisions must preserve the precedence relations defined by the DAG.

To address the CALBP-TS effectively, this study aims to answer the following key research questions:

- RQ1: How can the CALBP-TS be formally modeled to capture its unique characteristics (closed-loop topology, station revisitation, fixed-position machines, collaborative task execution) and high-dimensional complexity?

- RQ2: How can the combinatorial complexity arising from task-splitting (specifically, the non-linear workload allocation among multiple workers) be effectively resolved within an exact optimization framework, particularly in the master problem?

- RQ3: How can the computational efficiency of the classical Benders decomposition approach be significantly enhanced to solve large-scale CALBP-TS instances within practical time limits, especially through intelligent feasibility checks and cut generation?

Driven by these research questions, we propose a Benders decomposition framework that partitions the problem into a Master Problem (MP) and a Subproblem (SP). The MP optimizes a subset of assignment variables (e.g., process-worker or process-station allocations). The SP then resolves the remaining variables using fixed MP solutions. If infeasible, the SP generates Benders cuts to refine the MP model. This iterative procedure continues until infeasibility is confirmed or an optimal solution is found. The methodology originates from Hooker's seminal work [12] and aligns with recent applications in complex combinatorial optimization [4]. The key contributions of the study are listed below:

- From a theoretical standpoint, this paper introduces, for the first time, the Circular Assembly Line Balancing Problem with Task-Splitting (CALBP-TS). To tackle this complex multi-dimensional spatio-temporal assignment problem,

 

we propose an innovative integrated framework based on combinatorial Benders decomposition, decomposing the original problem into a master problem and s subproblem. Driven by the sequential logic of (worker-process) versus (process-station) decisions, two distinct decomposition strategies are presented: a top-down (worker-process priority) strategy and a bottom-up (process-station priority) strategy.

- Regarding the methodology, we devise a mathematical model and integrated framework for CALBP-TS, incorporating task-splitting into an exact algorithm via Benders-based decomposition. Task splitting, where multiple workers share a process, introduces a critical modeling challenge for calculating individual worker workloads within the master problem. This challenge is successfully resolved with rigorous theoretical proofs. For the subproblems, we develop a dummy process encoding technique. This technique circumvents the modeling complexity of task splitting within the subproblems by extending the process encoding scheme, thereby systematically addressing the end-to-end modeling challenge of task-splitting.

- From a managerial perspective, a Heuristic Infeasibility Proof (HIP) method to effectively detect infeasible solutions of the master problem is proposed. Furthermore, an Enhanced Benders Cut (EBC) generation algorithm is designed, which leverages information from infeasible master problem solutions to efficiently refine the model. The synergistic integration of these two strategies significantly improves the computational efficiency of the classical Benders decomposition approach.

The remaining sections of the study are as follows. Section 2 presents an extensive literature review of CALBP-TS. In Section 3, the problem definition, optimization model, numerical example, and problem complexity are given. Section 4 details the proposed algorithms. Section 5 describes the computational experiments performed through a design of experiment setting. Section 6 provides the concluding remarks, along with recommendations for future studies.

## 2 Literature review

Assembly lines are production systems widely applied to manufacturing industries with a high-volume output of standardised products. Their product-oriented layouts are generally built to fit flow-shops, which conveniently enables the mass production of homogeneous goods [13]. The Simple Assembly Line Balancing Problem (SALBP), introduced by Baybars [14], consists of assigning a set of tasks to workstations in such a way that precedence constraints are fulfilled, the time of each workstation does not exceed the cycle time and a given objective is optimized. There are two versions of the SALBP. In the first version (SALBP-1), cycle time is known, aiming to create a balanced line with the least number of stations. In the second version (SALBP-2), the number of stations to be opened is fixed, and the cycle time is attempted to be minimised [15]. Baybars (1986) [14] surveyed exact algorithms for SALBP, advancing the field by synthesizing key formulations, computational insights, and algorithmic efficiencies, while highlighting NP-hard challenges and future research gaps. Since then, research on the ALBP and its variants has significantly increased, covering areas such as multi-worker collaboration, machine assistance, human-robot collaboration, diverse line configurations, and multidimensional decision variables. This paper systematically reviews recent research advances in the Assembly Line Balancing Problem (ALBP), with relevant literature summarized in Table 1.

Straight-line configurations remain the dominant focus in ALBP research, with extensive literature dedicated to their optimization. The Multi-manned Assembly Line Balancing Problem (MALBP), prevalent in large-product manufacturing (e.g., automotive industries), extends classical ALBP by allowing multiple workers to perform tasks simultaneously at a single station, thereby reducing line length and labor costs. Prior exact methods for MALBP struggled with scalability, limiting solutions to small instances (≤45 tasks). Michels et al. (2019) [16] bridge this gap via a novel Benders decomposition algorithm with combinatorial cuts, which decomposes MALBP into a master problem and feasibility-seeking integer slave subproblems, outperforming prior exact methods on large-scale cases. Cao et al. (2020) [17] addressed ALBP with Uncertain Cycle Time (ALBP-UCT) using interval-based modeling and a multi-population genetic algorithm (MP-GA), minimizing

**Table 1. Summary of the literature review on ALBP/UALBP.**

| Author(s) & Year | Problem Type | Algorithm | Objectives to Minimize | Gaps (Fails to consider) |
|---|---|---|---|---|
| Baybars, I. (1986) [14] | SALBP | IP,B&B,DP | number of stations cycle time | general ALBP variant scalable solutions for large-scale problems (>50 tasks) |
| Michels et al. (2019) [16] | MALBP | BD | total workers number of stations | mixed-model lines zoning constraints scalability issues |
| Cao, Y. (2020) [17] | ALBP-UCT | MP-GA | number of stations cycle time ratio | real-world uncertainty additional constraints computational challenges |
| Jirasirilerd et al.(2020) [18] | SALBP-2 | VaNSAS | cycle time | employee skills, machines |
| Nourmohammadi et al. (2022) [19] | ALBP-HRC | MILP & SA | cycle time number of operators | cost-oriented objectives U-shaped or parallel lines |
| Andreu-Casas et al. (2022) [20] | MALBP | HEUR_PART | workers stations | U-shaped lines resource constraints stochastic task times |
| Katiraee et al. (2023) | ALWARBP | ε-constraint + CPLEX | cycle time task assignments | reassignment costs objective ergonomic metrics |
| Huang et al. (2024) | HRCALBP-II | Enhanced MIP + ICBD | cycle time | ergonomic risks cost optimization |
| Michels & Costa (2024) [21] | MALWIBP | MILP + HDHPS | total workers hierarchical stations | U-shaped/parallel layouts mixed-model production worker collaboration costs dynamic demand uncertainty |
| Nur et al. (2025) [22] | Stochastic ALBP | ILP+ heuristic | idle time total time | real-world validation dynamic task allocation |
| Schäfer et al. (2025) [23] | Complex ALBP | MILP | costs, area tolerance deviations cycle time | dynamic uncertainties buffer capacity optimization stochastic demand variability |
| Yilmaz(2022) | UALBP | AUGMECON2 | operational cost workload imbalance | demand uncertainty dynamic task times |
| Yilmaz et al. (2020) | UALBP | Robust Optimization | number of stations | cost/workload objectives demand dynamics |
| Huang et al. (2021) [24] | MTALBP-I | CBD | number of mated-stations | zoning constraints scalability for very large instances |
| Mao et al. (2023) | UALBP-HRC | EMIP + ESA | cycle time | mixed-model scenarios dynamic environments |
| This study | CALBP-TS | MILP+CBD HIP + EBC | workload imbalance | other economic objectives |

stations and cycle time while incorporating spatial/incompatible constraints and operator skill levels, enhancing adapt-ability to demand fluctuations. Jirasirilerd et al. (2020) [18] introduced a variable neighborhood strategy adaptive search method (VaNSAS) for SALBP-2 with machine constraints, minimizing cycle time in a garment industry case study, outper-forming existing methods. Nourmohammadi et al. (2022) [19] advanced ALBP research by developing a Mixed-Integer Linear Programming (MILP) model and adaptive Simulated Annealing (SA) for human-robot collaboration, optimizing cycle time and operator count while incorporating multi-operator stations and joint tasks. Andreu-Casas et al. (2022) [20] advanced multi-manned ALBP by addressing task-time dependencies from worker interference. It proposes HEUR_PART, a partition-based heuristic minimizing workers (primary) and stations (secondary), outperforming prior methods com-putationally, but it didn't address U-shaped layouts, resource limitations (tools/machinery), or stochastic time variability.

Katiraee et al. (2023) integrated workers' expertise and Borg-scale physical effort into assembly line rebalancing, enabling trainer-assisted task sharing. However, it neglects reassignment cost quantification and relies solely on subjective ergonomic assessment, omitting objective metrics like OCRA or energy expenditure. Huang et al. (2024) introduced an Enhanced Mixed-Integer Programming (MIP) model coupled with an Improved Combinatorial Benders Decomposition (ICBD) algorithm for human-robot collaborative assembly line balancing. Their approach achieved 100% feasibility and approximately 66% optimality with significantly reduced computational time. However, the study overlooks ergonomic factors and cost-efficiency considerations, limiting its applicability to real-world industrial settings where these objectives are critical. Michels & Costa (2024) [21] proposed and studied the problem of balancing assembly lines with multi-manned stations while integrating a heterogeneous workforce, specifically known as Multi-manned Assembly Line Worker Assignment and Integration Problem (MALWIBP). They proposed a Hierarchical Decomposition Heuristic with Proximity Search (HDHPS) that builds up on the multiple layers of decisions in the problem, but omit U-lines and mixed-model scenarios. Nur et al. (2025) [22] integrated stochastic task times and defect probabilities into ALBP via adjusted processing times, demonstrating efficiency gains. However, the framework neglects worker fatigue effects, relies on simulated data, and omits dynamic task allocation, limiting practical applicability. Schäfer et al. (2025) [23] proposed a Complex ALBP model integrating multi-criteria optimization, multi-robotic stations, and non-discrete task assignment (e.g., split welding). Their MILP approach with Gurobi reduces costs/space by 10.6%/43.4% in automotive assembly. However, it omits dynamic uncertainties (e.g., disruptions) and buffer design, limiting real-time adaptability.

In contrast, U-shaped layouts have gained significant traction in recent studies, offering enhanced flexibility for modern production systems. Yilmaz(2022) pioneered the integration of bi-objective U-shaped ALBP and parts feeding with worker heterogeneity, optimizing operational cost and workload imbalance via exact AUGMECON2. However, it fails to address demand uncertainty or dynamic task times, limiting practical adaptability in stochastic environments. In the same year, Yilmaz (2020) pioneered a robust optimization framework for U-shaped ALBP with uncertain task times, integrating worker assignment via interval-polyhedral uncertainty sets. However, it overlooks multi-objective trade-offs (e.g., cost/ workload imbalance) and dynamic demand fluctuations, limiting holistic applicability. Huang et al. (2021) [24] proposed a combinatorial Benders decomposition algorithm with sequence-based cut generation for the mixed-model two-sided ALBP (MTALBP-I), achieving exact solutions for large instances (≤148 tasks). However, it fails to incorporate zoning constraints and struggles with very large problems (e.g., P205 instances). Mao et al. (2023) first addressed the U-type assembly line balancing problem with collaborative robots (UALBP-HRC) enabling parallel and collaborative tasks. They proposed an enhanced MIP model and simulated annealing algorithm (ESA) to minimize cycle time, significantly reducing gaps versus benchmarks. However, the study fails to consider mixed-model production or dynamic task reassignment.

This study pioneers the integration of circular assembly lines into the ALBP framework by proposing a novel Circular Assembly Line Balancing Problem with Task Splitting (CALBP-TS), which enables processes re-entry, stochastic worker-to-station assignment, and multi-stage processing with hybrid fixed/mobile machinery through dynamically coordinated mobile machine and workers. Subject to maximum circle count constraints and operational limitations, these features substantially escalate spatio-temporal constraint complexity. To address this, we develop a Combinatorial Benders Decomposition (CBD) framework that decomposes the primal problem into two subsystems while innovatively integrating Heuristic Infeasibility Proof (HIP), Enhanced Benders Cuts (EBC), and branching strategies, collectively compressing the search space and accelerating convergence to near-optimal solutions.

The proposed Enhanced Benders Decomposition (EBD) algorithm can be directly adapted to disassembly-line balancing (DLBP) and seru scheduling problems, as these domains share core challenges with CALBP-TS: high-dimensional resource allocation, spatio-temporal constraints, and collaborative task execution. For DLBP with multi-manned stations and uncertain task times (e.g., Yeni et al., Comput. Ind. Eng., 2024 [25]), the EBD framework would decompose the problem into: (1) a master problem assigning disassembly tasks to workers while optimizing workload balance (using our linearization theorem to handle shared tasks), and (2) a subproblem enforcing sequence-dependent constraints via

dummy station encoding technique (§3.3.2), with Enhanced Benders Cuts (EBC) resolving spatial conflicts for fixed tools/robots (Huang et al., J. Manuf. Syst., 2025 [26]). The Heuristic Infeasibility Proof (HIP) would accelerate feasibility checks for station revisitation, critical in partial disassembly cycles. Similarly, for seru scheduling with lot streaming and worker transfers (Gürsoy Yılmaz et al., Comput. Ind. Eng., 2023 [27]; Comput. Oper. Res., 2025 [28]), the master problem would optimize task splitting for batch assignments across seru cells, while the subproblem models worker mobility constraints using adjacency-based dummy processes. Local Branching (§4.3) would dynamically adjust transfers to minimize movement penalties, and HIP-EBC synergy (empirically reducing runtime by 13.7% in CALBP-TS) would prune solutions violating skill-cell compatibility. Critically, EBD's modularity accommodates stochastic demand (Nur et al., 2025 [22]) via robust MP formulations, making it a versatile framework for reconfigurable systems. Future work will implement EBD for these domains, leveraging its scalability for large-scale instances.

## 3 Problem definition and optimization model

### 3.1 Problem definition

As a variant of the ALBP, CALBP-TS finds broad applicability across diverse manufacturing sectors, including automotive assembly, semiconductor fabrication, and textile production. Fundamentally, line balancing strategically partitions and sequences operations according to engineering workflows to equalize workstation loads, thereby maximizing production efficiency. Crucially, effective line balancing necessitates the simultaneous optimization of two key aspects:: (i) process-station-machine assignments, and (ii) inter-station workload distribution. This dual-focus approach enhances overall line balance integrity while driving systemic productivity gains.

A defining characteristic of a circular production line is its allowance for processes to re-enter the same station multiple times. As depicted in Fig 2, where stations are sequentially ordered 1, 2,..., 6, and processes 1–7 must be completed strictly in sequence, the illustrated allocation on the right necessitates that the process bypasses station 1 after operation 6 until operation 7 requires re-entry into station 1 during a subsequent cycle. This constitutes one repeated station entry at station 1. Minimizing such repeated entries is a critical production objective.

While most processes utilize mobile standard machine, certain processes require fixed-location machine anchored to specific stations. This machine cannot be relocated or added. Fig 3 exemplifies this constraint: if the machine for process 2 is fixed exclusively at station 3, the process, after completing process 1 at station 1, must bypass station 2 and proceed directly to station 3 for process 2. Consequently, given the finite number of stations, worker F can only be assigned to station 2. Process 6 is then completed by worker F in the second cycle, while process 7 is executed by worker A in the third cycle.

Another defining feature of CALBP-TS, distinguishing it from prior approaches, is its explicit incorporation of process task splitting. This mechanism allows a single set of processes to be partitioned among multiple workers, enabling shared workload distribution. Consequently, CALBP-TS achieves finer-grained regulation of worker workloads. This operational

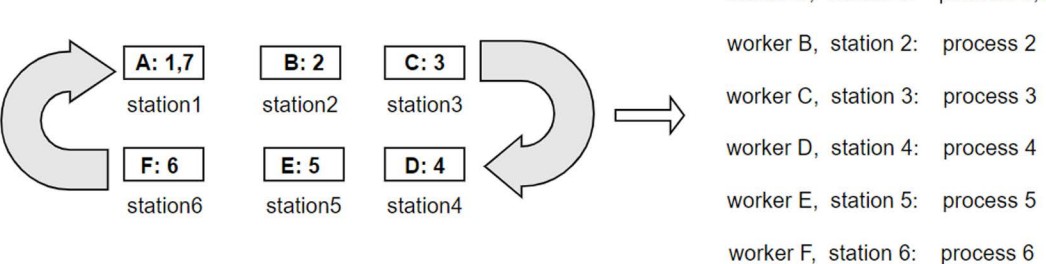

**Fig 2. Schematic diagram of process assignment results without machine.**

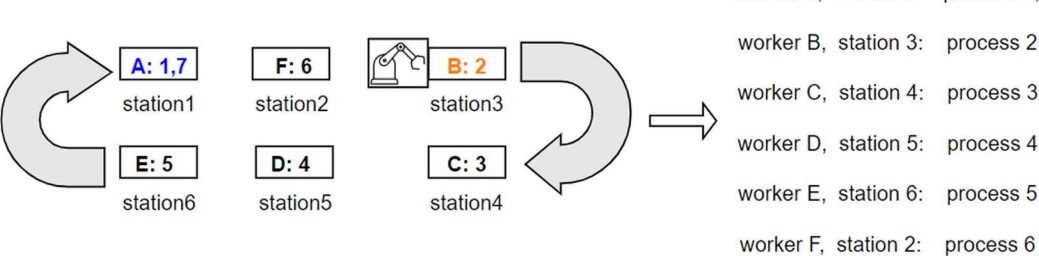

**Fig 3. Schematic diagram of process assignment results with machine.**

flexibility, however, introduces significant allocation complexity: workers may be assigned to multiple stations across different cycles, and multiple workers may perform distinct processes at the same station during different cycles. As illustrated in Fig 4, Workers 1 and 2 collaboratively process the process set {1,2,6} through dynamic workload sharing, where Worker 2 alleviates the workload of Worker 1. This set comprises sequentially constrained processes ($1 \to 2 \to 6$), with Process 2 requiring fixed-location machine. The allocation scheme yields a cycle-specific execution pattern: during Cycle 1, Process 1 is concurrently executed by Worker 1 at Station 1 and Worker 2 at Station 3, while Process 2 is simultaneously performed by Worker 1 at Station 7 and Worker 2 at Station 5; in Cycle 2, Process 6 is jointly completed by Worker 1 at Station 2 and Worker 2 at Station 4. This strategy optimizes resource utilization through coordinated deployment of worker and machine, thus significantly reducing total processing time while enhancing inter-worker workload balancing.

In summary, CALBP-TS is a multi-dimensional resource balanced allocation problem with rather complex constraints. Its objective functions are to minimize the maximum workload and minimize the workload gap among workers, to address the issues of "work overload" and "work idleness," thereby improving production efficiency. To enhance conceptual clarity of the CALBP-TS, Table 2 systematically synthesizes the optimization objectives and a five-dimensional constraint framework comprising: (1) Process constraints, (2) Worker constraints, (3) Machine constraints, (4) Station constraints, and (5) Domain-specific constraints.

## 3.2 Optimization model

In this section, a decomposition strategy based on Combinatorial Benders Decomposition is proposed. Building upon this strategy, the original Mixed-Integer Linear Programming (MILP) problem is decomposed into two subproblems: a Master Problem incorporating the optimization objective, and a Feasibility Subproblem dedicated solely to constraint handling.

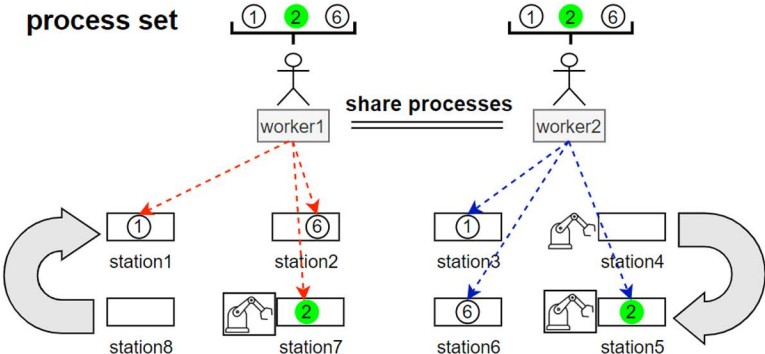

**Fig 4. Schematic diagram of process assignment with task-splitting.**

**Table 2. Optimization objectives and multidimensional constraints for the CALBP-TS.**

| Type | Objectives and Constraints |
|---|---|
| Optimization Objectives | Minimize the maximum workload and the workload volatility among workers. |
| Process Constraints | 1. Operations with strict precedence constraints shall retain their sequential workflow. |
| | 2. A maximum of *max_split_num* operation bundles per process may be split among workers. |
| | 3. Batch production requires single process/bundles to be assigned to one worker, except for bottleneck operations exceeding time thresholds. Such processes may be distributed to up to *max_worker_per_oper* workers (maximum workers per process). Example: With *max_worker_per_oper*=2, 1000 units can be split as 500 units to Worker A and 500 to Worker B. |
| Worker Constraints | 1. All available workers must be assigned to processes. |
| | 2. Workers must possess required skills for assigned processes. |
| | 3. Worker capacity is limited to *max_station_per_worker* stations. |
| | 4. Functionally identical processes require workers with exact skill matches. If unavailable, workers with category skills may be assigned. |
| | 5. Specific processes may be pre-assigned to designated skilled workers, with or without station specification. |
| Machine Constraints | 1.Station space limits machine to *max_machine_per_station* units. |
| | 2. Large-scale machine exclusively occupies entire stations. |
| | 3. Processes requiring identical machine types but different specifications (e.g., "m-688067-01" with subtypes "A"/"B") necessitate separate machines. |
| | 4. Fixed machine positions are immutable. |
| Station Constraints | 1.Each station accommodates at most one worker. |
| | 2. Adjacent stations in circular lines include radially opposite stations within accessible proximity. |
| Other Constraints | 1. Processes cycling is capped at *max_cycle_count* cycles. |
| | 2. Workload balance: All workers' operation times(workload) shall deviate within ±*volatility_rate*% of the mean workload. |
| | 3. Total station revisit events shall not exceed *max_revisited_station_count* (exempting fixed-machine stations; others ≤2 revisits for each station). |

**3.2.1 Decomposition method.** As illustrated in Fig 1 and the preceding description of the CALBP-TS problem, CALBP-TS constitutes a multi-dimensional resource allocation problem involving the assignment of processes, workers, stations, and machines. Such problems are typically addressed by formulating a complete MILP model and solving it using solvers like Gurobi. However, the CALBP-TS problem studied in this paper is a complex combinatorial optimization problem where solving the full MILP model directly proves intractable within limited computational resources. Consequently, we employ the combinatorial Benders decomposition method for mixed integer programming, first proposed by Codato and Fischetti [29], with the aim of removing the model dependency on the big-M coefficients.

Driven by the objective of worker load balancing, we decompose the problem into two components: The Master Problem (MP), an optimization problem, addresses process-worker assignment, task splitting, and optimizing the worker load balancing objective; whereas the Subproblem (SP), a feasibility problem, manages process-station assignment, machine-station assignment, and enforces complex constraints including the maximum cycle count, precedence constraints for split tasks, and station allocation requirements. The SP functions as a constraint satisfaction model without an explicit optimization objective. As shown in Fig 5.

As shown in Fig 6, based on the sequence of solving these subproblems, we designed two decomposition strategies. The Top-Down strategy first solves process-worker assignment followed by process-station-machine assignment, this strategy is applicable to instances characterized by simple inter-process dependencies. In contrast, the Bottom-Up strategy first solves process-station-machine assignment followed by process-worker assignment, this strategy is suitable for instances with complex inter-process dependencies.

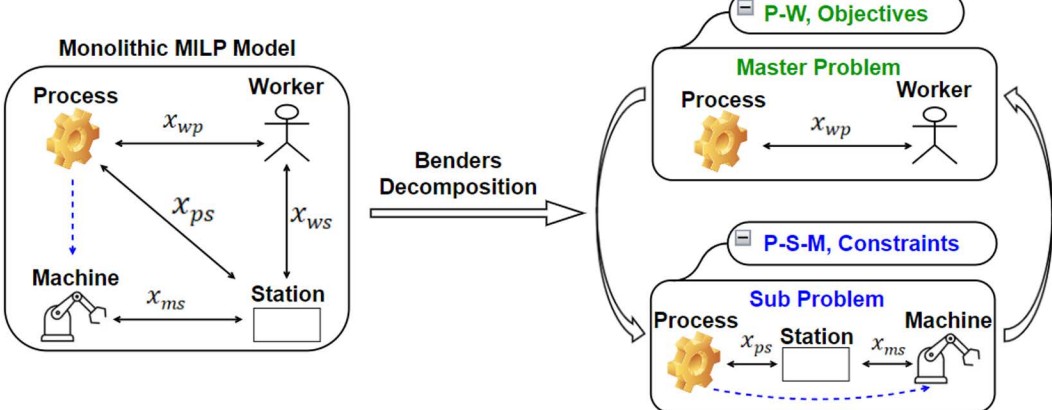

**Fig 5. Schematic diagram of decomposition from monolithic MILP Model to two sub models.**

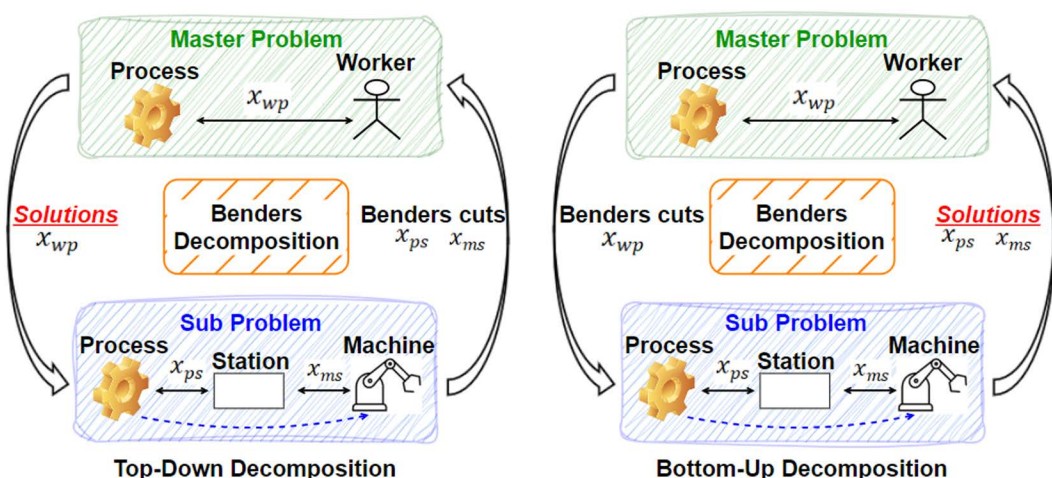

**Fig 6. Top-down vs. bottom-up benders decomposition strategies.**

### 3.2.2 Master problem model (top-down approach).

As mentioned above, the objective of the master problem is to address the worker-to-process allocation, considering task splitting and workload volatility constraints. The specific mathematical modeling is detailed below.

Sets

 $W$: Workers – indexed by $w$

 $P$: Processes – indexed by $p$

Parameters

 $standTime_p$: Stand operation time of process $p$

 $effi_{wp}$: The efficiency of worker $w$ while doing process $p$

 $skillCap_{wp}$: Whether worker $w$ possessing the skill of doing process $p$

 $catCap_{wp}$: Whether worker $w$ possessing the skill category of doing process $p$

 $W_p$: Worker set with required skills of doing process $p$

$P_w$: Process set that worker $w$ is capable of processing

$\overline{w}_p$: Pre-specified assignment of worker w to process $p$

*volRate*: The worker time volatility rate of workers

*maxW$_p$O*: Maximum number of workers for each process

*maxS$_p$O*: Maximum number of stations for each process

*maxSplit*: Maximum number of task split for each process

Decision Variables

$x_{wp}$: 1, if worker w is assigned to process $p$; 0 otherwise ($\forall w \in W, p \in P$)

$y_w$: the rhythm of worker $w$ ($\forall w \in W$)

Auxiliary Variables (with Task-Splitting)

$x'_{wp}$: denoting the workload coefficient of worker $w$ on process $p$, ($\forall w \in W, p \in P$)

$h_{wpw'}$: $x'_{wp}$, if worker $w'$ is assigned to process $p$, 0 otherwise ($\forall w, w' \in W, p \in P$)

Objectives

$$\text{Minimize} \sum_{w_1, w_2 \in W} \left| y_{w_1} - y_{w_2} \right| \tag{1}$$

Constraints (without Task-Splitting)

$$y_w = \sum_{p \in P} \frac{standTime_p}{effi_{wp}} \cdot x_{wp}, \forall w \in W \tag{2}$$

$$\sum_{w \in W} x_{wp} = 1, \forall p \in P \tag{3}$$

$$\sum_{w \in W_p} x_{wp} = 1; \sum_{w \in W - W_p} x_{wp} = 0, \forall w \in W, W_p \neq \varnothing \tag{4}$$

$$\sum_{p \in P} x_{wp} \geq 1, \forall w \in W \tag{5}$$

$$\sum_{w \in W} x_{wp} \leq \max(\sum_{w \in W} skillCap_{wp}, \sum_{w \in W} catCap_{wp}), \forall p \in P \tag{6}$$

$$x_{\overline{w}_p p} = 1; x_{wp} = 0, \forall p \in P, w \in W - \{\overline{w}_p\} \tag{7}$$

$$\begin{cases} \overline{y} = \frac{\sum_{w \in W} y_w}{|W|} \\ \overline{y} \cdot (1 - volRate) \leq y_w \leq \overline{y} \cdot (1 + volRate) \end{cases}, \forall w \in W \tag{8}$$

$$x_{wp} \in \{0, 1\}, \ y_w \in \mathbb{R}^+, \forall w \in W, p \in P \tag{9}$$

Constraints (with Task-Splitting)

$$\begin{cases} y_w = \sum_{p \in P} \frac{standTime_p}{effi_{wp}} \cdot \frac{x_{wp}}{\sum_{w \in W} x_{wp}} (original\ formulation) \\ y_w = \sum_{p \in P} \frac{standTime_p}{effi_{wp}} \cdot x'_{wp} (linearized\ formulation) \end{cases}, \forall w \in W \tag{10}$$

$$\begin{cases} h_{wpw'} \geq -1 \cdot x_{w'p} \\ h_{wpw'} \leq x_{w'p} \end{cases}, \forall w, w' \in W, p \in P \tag{11}$$

$$\begin{cases} h_{wpw'} - x'_{wp} \geq -1 \cdot (1 - x_{w'p}) \\ h_{wpw'} - x'_{wp} \leq 1 - x_{w'p} \end{cases}, \forall w, w' \in W, p \in P \tag{12}$$

$$x_{wp} = \sum_{w' \in W_p} h_{wpw'}, \forall p \in P \tag{13}$$

$$1 \leq \sum_{w \in W} x_{wp} \leq max(maxW_p O, maxS_p O), \forall p \in P \tag{14}$$

$$\sum_{w \in W_p} x_{wp} \geq 1; \sum_{w \in W - W_p} x_{wp} = 0, \forall w \in W, W_p \neq \varnothing \tag{15}$$

$$x'_{wp} \subseteq [0, 1], h_{wpw'} \subseteq [0, 1], \forall w, w' \in W, p \in P \tag{16}$$

The optimization objective of the model is formulated to minimize the sum of absolute differences in workload between all pairs of workers, as specified in Equation (1). Constraints governing the scenario without task splitting are given by Equations (2)-(9). Equation (2) computes the workload for each worker. Crucially, for any given process, a worker's workload decreases with higher efficiency. Equation (3) enforces the exclusive assignment of each process to a single worker. Equation (4) restricts candidate assignments for a process to workers possessing the requisite skill. Equation (5) allows workers to be assigned to multiple processes. Equation (6) limits the maximum number of workers assignable to a process to the number of workers qualified to perform it. Equation (7) incorporates pre-defined assignments. Equation (8) bounds the allowable workload fluctuation per worker.

Incorporating task splitting necessitates constraint modifications. Workload calculation must now account for individual worker proportions. Equation (10) presents both the original and linearized formulations for workload under splitting. To enable linear computation, auxiliary variables are introduced: $x'_{wp}$ (denoting a worker's proportional share) and $h_{wpw'}$. The variable $h_{wpw'}$ represents $x'_{wp}$ specifically when worker $w'$ is assigned to process $p$. The linearization is achieved via Equation (11)-(13). Equation (11) forces $h_{wpw'} = 0$ if assignment $x_{w'p} = 0$, Equation (12) sets $h_{wpw'} = x'_{wp}$ if $x_{w'p} = 1$. Consequently, we can come to the conclusion that $h_{wpw'} = x'_{wp} \cdot x_{w'p}$. Collectively, Equation (11) and (12) establish the functional link between the binary decision variable $x_{w'p}$ and the continuous auxiliary variables $x'_{wp}$ and $h_{wpw'}$. Equation (13) ensures that if worker $w$ is assigned to process $p$, they share an equal workload proportion with all other workers assigned to $p$. This elegant linearization technique effectively resolves the nonlinearity inherent in calculating workload proportions

under task splitting (due to heterogeneous worker efficiencies) and in the original objective function. Moreover, it successfully decouples assignment relationships from continuous variables, a critical capability for practical industrial scheduling applications. Detailed numerical example is provided in Section 3.3.

Equation (14) constrains the maximum number of workers assignable to a process based on skill proficiency limitations and maximum workstation capacity. Equation (15) mandates that assigned workers must be selected from the pool with the required skill, and allows multiple assignments (>1). The domains of all variables are defined by Equations (9) and (16).

### 3.2.3 Subproblem model (top-down approach).
The subproblem model must address station assignments. Given the inherent cyclic constraints in practical circular production lines, direct modeling would incur prohibitive complexity. To mitigate this, we introduce the maximum cycle parameter maxCycle to project the original station $S'$ onto a virtually expanded station $S$. This transformation logically converts the circular structure into a linear configuration, substantially simplifying subproblem formulation. As illustrated in Fig 7, assuming original stations are numbered sequentially from 1, the initial configuration comprises 8 stations. With a maximum circle constraint set to 5, the station indices are expanded from 1−8−1−40 through cyclic numbering. Blue-labeled indices (1, 9, 17, 25, 33) correspond to same circle-specific station (circle 1 to circle 5). The mapping relationship follows: $S'\_id = (S\_id − 1) \bmod |S'| + 1$.

Sets

$S'$: Original Stations – indexed by $s'$

$\overline{S}'$: Stations with fixed machines

$C$: Cycle count – indexed by $c$

$M$: Machines – indexed by $m$

$monoM$: Monopolistic Machine – indexed by $m$

$procFM$: Process with fixed machine – indexed by $(p, m)$

$S = S' \cdot maxCycle$: Expanded Station – indexed by $s$

$PP$: Process Precedence – indexed by $(i, j)$

Parameters

$\overline{x}_{wp}$: Worker-process assignment from master problem

$maxS_pW$: Maximum number of stations per worker

$maxM_pS$: Maximum number of machines of stations

$maxRS$: Maximum number of revisit count of stations

$maxCycle$: Maximum cycle count

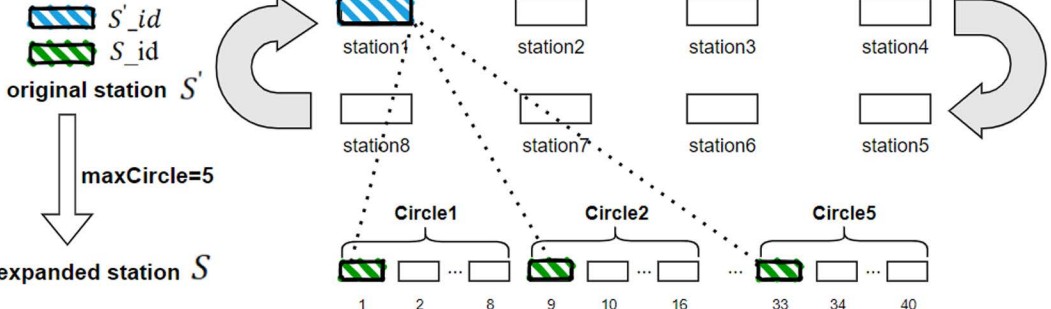

**Fig 7. Layout transition diagram of production line stations: circular to straight configuration.**

Decision Variables

$x'_{ps}$: 1, if process $p$ is assigned to original station $s$; 0 otherwise ($\forall p \in P, s \in S'$)

$x_{ps}$: 1, if process $p$ is assigned to expanded station $s$; 0 otherwise

($\forall p \in P, s \in S, s$ is the expanded station of the original station $s'$)

$y_{ws}$: 1, if worker $w$ is assigned to original station $s$; 0 otherwise ($\forall p \in P, s \in S'$)

$z_{ms}$: 1, if machine $m$ is assigned to original station $s$; 0 otherwise ($\forall m \in M, s \in S'$)

$v_{cs}$: 1, if original station $s$ is assigned within circle $c$; 0 otherwise ($\forall s \in S', c \in C$)

$r_s$: process revisit count for original station $s$ ($\forall s \in S'$)

Variable Implication

$$x_{ps} \Rightarrow x'_{ps'}, \ \forall p \in P, s \in S, s' \in S' \tag{17}$$

$$\overline{x}_{wp} \cdot x'_{ps} \Rightarrow y_{ws}, \ \forall w \in W, p \in P, s \in S' \tag{18}$$

$$x_{p,s+c \cdot |S'|} \Rightarrow v_{cs}, \ \forall p \in P, s \in S', c \in C \tag{19}$$

Constraints

$$\sum_{s \in S} x_{ps} = 1, \ \forall p \in P \tag{20}$$

$$\sum_{s \in S'} x_{p_1 s} \cdot s \leq \sum_{s \in S'} x_{p_2 s} \cdot s, \ \forall p_1, p_2 \in PP \tag{21}$$

$$r_s = max(0, \sum_{c \in C} v_{cs} - 1), \ \forall s \in S' \tag{22}$$

$$r_s \leq 2, \ \forall s \in S' - \overline{S}' \tag{23}$$

$$\sum_{s \in S'} r_s \leq maxRS \tag{24}$$

$$x'_{ps} \leq z_{ms}, \forall s \in S', (p, m) \in procFM \tag{25}$$

$$\sum_{m \in M} z_{ms} \leq maxM_pS, \forall s \in S' \tag{26}$$

$$z_{m_1 s} + z_{m_2 s} \leq 1, \forall s \in S', m_1, m_2 \in monoM, m_1 \neq m_2 \tag{27}$$

$$x'_{ps} \in \{0, 1\}, \ x_{ps} \in \{0, 1\}, y_{ws} \in \{0, 1\}, \forall p \in P, s \in S', w \in W \tag{28}$$

$$v_{cs} \in \{0, 1\}, \forall c \in C, s \in S' \tag{29}$$

$$r_s \in Z_0^+, \forall s \in S' \tag{30}$$

The subproblem model focuses solely on constraint satisfaction without incorporating an explicit optimization objective. Equation (17) ensures that the occupation of an extended station implies the assignment of its corresponding original station. Equation (18) establishes the linkage between worker-process-station assignment variables: if worker $w$ is assigned to process $p$ and process $p$ is assigned to original station $s$, then worker $w$ must be assigned to station $s$. Equation (19) calculates the number of times station $s$ is used; the sequential numbering of extended stations allows determination of the specific cycle in which it is utilized. Equation (20) stipulates that each process must be assigned to exactly one station. Equation (21) enforces process precedence dependencies by requiring that the station number assigned to any predecessor process $p_1$ must be strictly less than the station number assigned to its successor process $p_2$. Equations (22)-(24) govern the logic for station re-entry, the constraint meaning of which is detailed in item 3 of the "Other Constraints" section in Table 2. Equation (25) specifies that when a process requiring a particular machine type is assigned to a station, that station must be equipped with the corresponding machine. Equation (26) limits the number of machines assigned to any station to its maximum capacity. Equation (27) enforces that exclusive machines can only be assigned to a single station, meaning no station can host two or more exclusive machines. Finally, Equations (28)-(30) define the domains of all decision variables.

### 3.3 Numerical example

This section employs numerical cases to elucidate the core challenges of the model and corresponding solution techniques. In the master problem model, task splitting effectively optimizes the objective function but introduces implementation challenges for linearizing the optimization objective. To address this, we propose a linearization technique that allocates workloads among workers sharing the same process set — a requirement driven by real-world Huawei data scenarios. While the subproblem formulation does not explicitly consider task splitting, which must be incorporated in the monolithic MILP model. By mapping multi-worker collaborative processes to virtual processes and assigning these during subproblem resolution, we elegantly resolve the complex process-to-station assignment constraints arising from task splitting.

**3.3.1 Linearization theorem proof and numerical example.** Theorem 1 (Equiproportional Workload Sharing).
Given Equations (11)-(13) in the linearized model, for any process $p$ and worker $w$ assigned to $p$ ($x_{wp}$=1):

$$x'_{wp} = \frac{1}{n_p} \qquad where \qquad n_p = \sum_{w' \in W_p} x_{w'p}$$

**Proof.**

From Equation (13): $x_{wp} = \sum_{w' \in W_p} h_{wpw'}$
Equations (11) and (12) enforce $h_{wpw'} = x'_{wp} \cdot x_{w'p}$
Substitution yields:

$$x_{wp} = \sum_{w' \in W_p} h_{wpw'} = \sum_{w' \in W_p} x'_{wp} \cdot x_{w'p} = x'_{wp} \cdot \sum_{w' \in W_p} x_{w'p} = x'_{wp} \cdot n_p$$

Thus $x'_{wp} = \frac{x_{wp}}{n_p}$. When $x_{wp}$=1, $x'_{wp} = \frac{1}{n_p}$.

**Proof End.**

Assume that three workers are allocated to process P. P: {A, B, C}, then we have $n_p$=3. From the above conclusion we can have the knowledge shown on Table 3.

**3.3.2 Dummy process coding numerical example.** When considering task splitting, a single process may be assigned to multiple workers in parallel (e.g., Process 5 in Fig 8 is simultaneously allocated to Workers 1, 2, and 3). This allocation raises three critical challenges in subproblem decision-making for process-to-station assignments:

(1) **Process Coding Conflict**: How to distinguish execution instances of the same process across different workers;

(2) **Sequential Consistency Maintenance**: How to preserve the original precedence constraints during parallel execution;

(3) **Constraint Invalidation**: The constraint in Equation (21) becomes invalid under task-splitting scenarios, necessitating a redesigned modeling framework.

To systematically resolve these complexities, we propose a dummy process coding technique. By eliminating direct dependencies on task-splitting logic in subproblem modeling, this approach significantly reduces model complexity while rigorously ensuring that allocation outcomes achieve functional equivalence with the original task-splitting requirements.

As illustrated in Fig 9, the dummy process coding technique operates through the following mechanism:

(1) **Initialization**: Set the starting code for dummy processes to the next natural number after the current process size parameter ($|P| + 1$);

(2) **Mapping Maintenance**: Establish a dictionary $P_{map}$ to map dummy processes to their original counterparts;

(3) **Dynamic Update Protocol**: For each task-splitting action, do

- Update $P_{map}$ to register new dummy-original process mappings
- Increment $|P|$ to expand the coding space
- Reconstruct the process precedence topology to preserve original sequential dependencies after dummy process insertion.

As demonstrated in Fig 10, the dummy process coding technique resolves all threechallenges by simply extending the process dataset prior to subproblem solving. Crucially, this approach eliminates the need for explicit handling of intricate task-splitting logics within the subproblem's mathematical formulation, thereby enhancing computational tractability and model scalability.

**Table 3. Numerical example of linearization formulation.**

| Worker Combination ($w$,$w'$) | $x_{wp}$ | $x'_{wp}$ | $h_{wpw'}$ | Explanation of $h_{wpw'}$ |
|---|---|---|---|---|
| (A,A) | 1 | 1/3 | 1/3 | $x'_{Ap}$, Worker A is assigned |
| (A,B) | 1 | 1/3 | 1/3 | $x'_{Ap}$, Worker B is assigned |
| (A,C) | 1 | 1/3 | 1/3 | $x'_{Ap}$, Worker C is assigned |
| (B,A) | 1 | 1/3 | 1/3 | $x'_{Bp}$, Worker A is assigned |
| (D,A) | 0 | 0 | 0 | Worker D is not assigned |
| (A,X) | 1 | 1/3 | 0 | Worker X is not assigned |

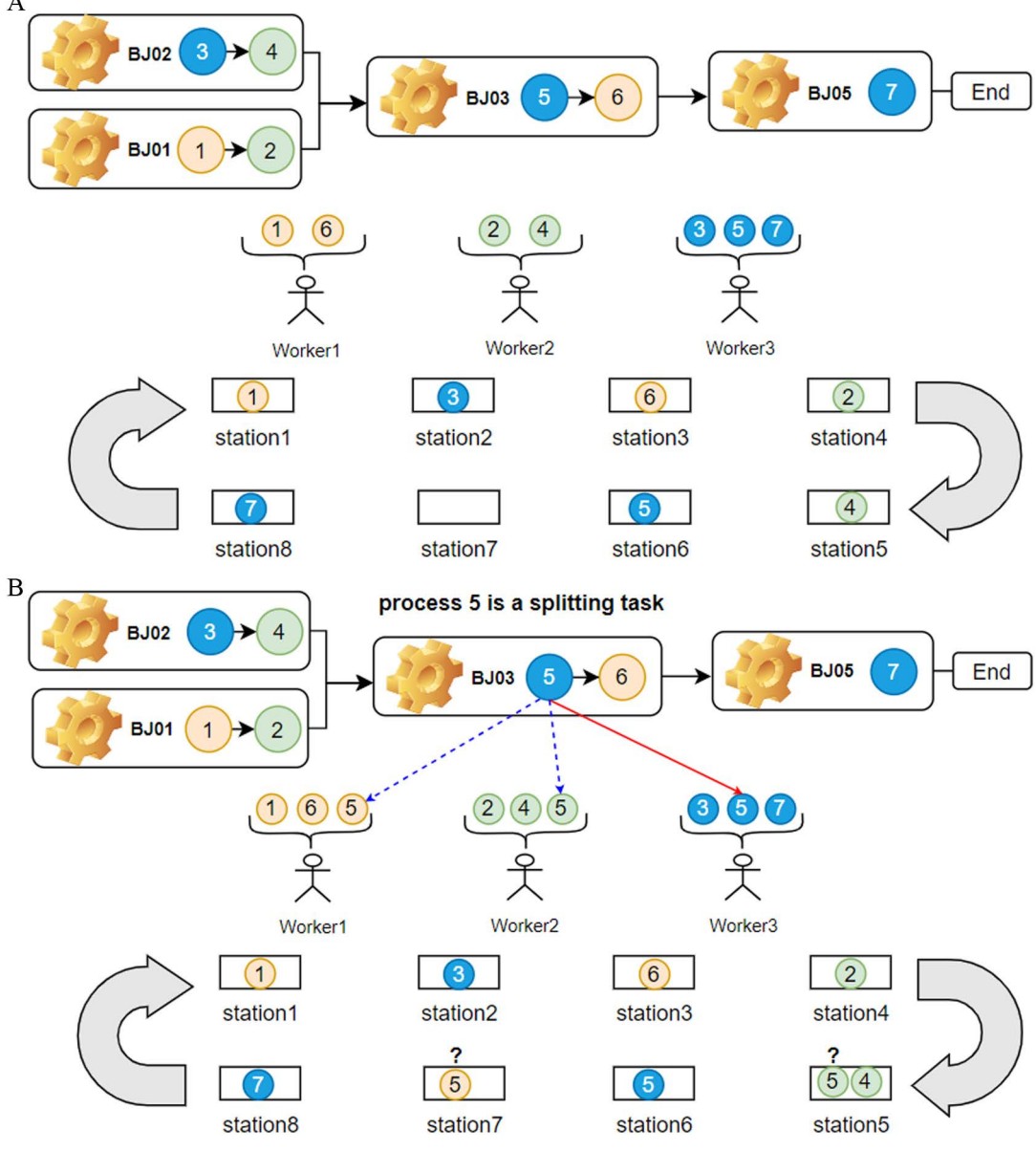

**Fig 8. Quantitative allocation chart for process-worker-station assignments.** (a) without Task-Splitting. (b) with Task-Splitting.

## 4 Proposed algorithms

### 4.1 Integrated algorithmic framework

As outlined in Section 3, we employ Combinatorial Benders Decomposition (CBD) to partition the original MILP problem into a master problem (MP) and a subproblem (SP), effectively addressing the spatio-temporal computational complexity inherent in CALBP-TS. This chapter presents an innovative algorithmic framework for coordinated master-subproblem solving, with two pivotal technical advancements: (i) Heuristic Infeasibility Proof (**HIP**) algorithm that rapidly verifies SP feasibility based on MP solutions, accelerating the overall solution process; (ii) Enhanced Benders Cut (**EBC**) generation procedure that dynamically derives strong cutting planes leveraging HIP results, significantly reducing the MP's search space.

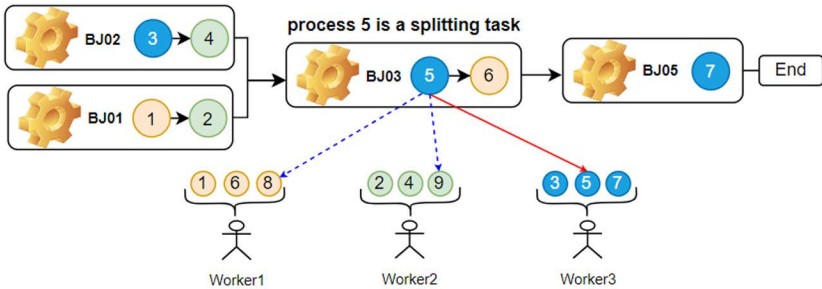

**Dummy Process Coding**

| W-P | P size | P map | Precedence |
|---|---|---|---|
| Worker1-{1,6,5} + Worker2-{2,4,5} | $|P|=7$ | {5:5} | {1:5, 2:5, 3:5, 4:5, 5:6, ...} |
| Worker1-{1,6,5} ⟹ Worker1-{1,6,**8**} | $|P|=8$ | {5:5, **8**:5} | {1:5, 1,**8**, 2:5, 2,**8**, 3:5, ...} |
| Worker2-{2,4,5} ⟹ Worker1-{2,4,**9**} | $|P|=9$ | {5:5, **8**:5, **9**:5} | {1:5, 1,**8**, 1,**9**, 2:5, 2,**8**, ...} |

**Fig 9. Schematic diagram of dummy process coding numerical demonstration.**

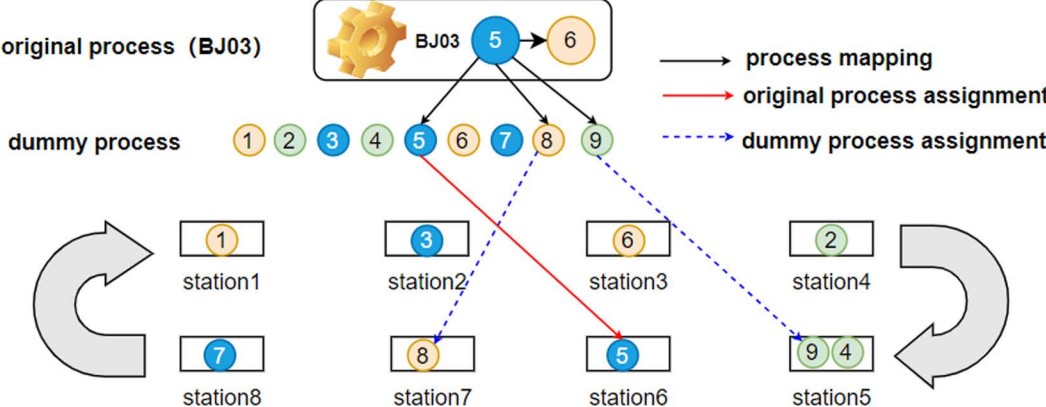

**Fig 10. Process-station assignment diagram incorporating dummy process coding.**

Fig 11 depicts the integrated algorithmic framework. Initially, we solve the MP without considering task splitting. If infeasible, we solve the MP again incorporating task splitting. This two-phase approach stems from empirical observation: some instances necessitate task splitting to achieve feasibility. Subsequently, before modeling the subproblem, we utilize the HIP algorithm to check the feasibility. If HIP deems the subproblem infeasible, we proceed directly to the Local Branching phase to adjust the model and re-validate the solution. Should no feasible solution remain after this adjustment, we add EBC to refine the MP model. This process iterates cyclically until either the computational time limit is reached or an optimal solution is successfully returned.

### 4.2 Heuristic infeasibility proof (HIP)

Topological sorting is the process of arranging all vertices in a directed graph into a linear sequence such that for any pair of vertices u and v, if there exists a directed edge <u,v> in the graph, then u precedes v in the linear sequence. As illustrated in Fig 12, given the process topology graph and the worker-process assignment solution derived from the master

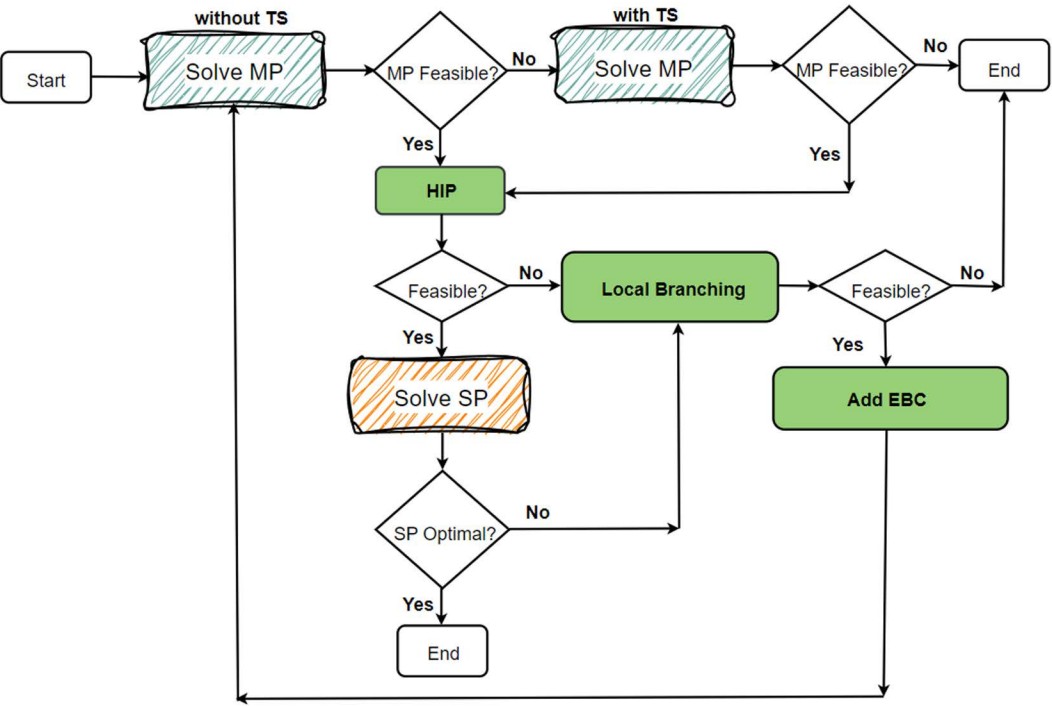

**Fig 11. Integrated algorithmic framework.**

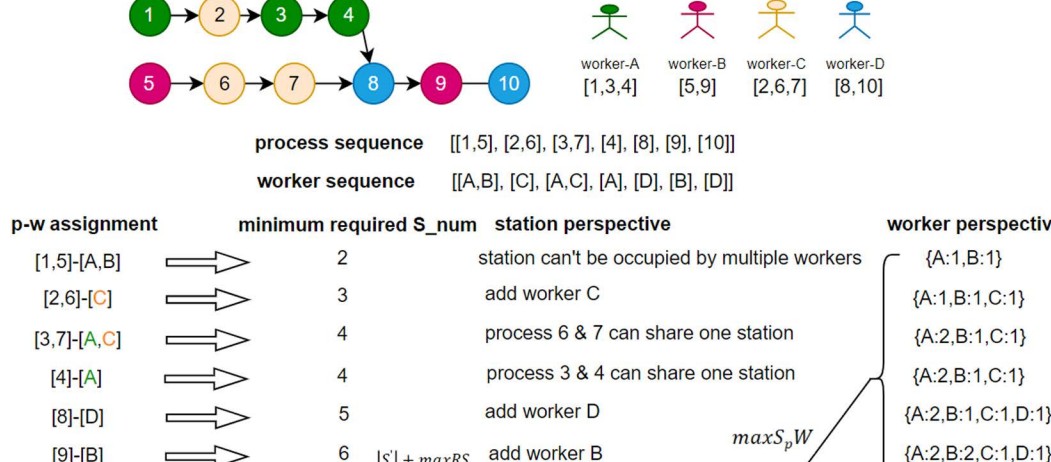

**Fig 12. Diagram illustrating the computational process of the HIP algorithm.**

problem, the minimum required number of stations $|P|$ and the corresponding worker-station mapping can be computed based on feasible processing sequences respecting precedence constraints (Fig 12 demonstrates one assignment scenario) and the associated worker sequence.

J.N. Hooker [12] has pointed out that Benders decomposition uses a strategy of "learning from one's mistakes" that has been employed in a more general way by constraint satisfaction methods. It uses this information to reduce the number of solutions it must enumerate to find an optimal solution. Building upon this principle, this paper proposes the Heuristic Infeasibility Proof (HIP) algorithm. HIP evaluates the feasibility of an assignment solution $\bar{x}_{wp}$ for the subproblem by comparing S_num with the maximum allowed stations $|S'| + maxRS$. If S_num exceeds $|S'| + maxRS$ or the solution inherently violates the $maxS_pW$ constraint (i.e., the number of distinct stations assigned to any worker surpasses $maxS_pW$), then solution $\bar{x}_{wp}$ is definitively infeasible for the subproblem. Consequently, modeling and solving the subproblem becomes unnecessary, and the algorithm proceeds directly to the next step, specifically the local_branching phase depicted in Fig 11.

The computational procedure in Fig 12 is detailed as follows. The notation [1,5]-[A,B] indicates that processes 1 and 5 (which have no precedence constraint between them) are assigned to workers A and B, respectively. This necessitates two distinct stations since a single station cannot be assigned to two different workers concurrently, resulting in the worker-station count hash table {A:1, B:1}. Subsequently, [2,6-[C] signifies processes 2 and 6 assigned to worker C. As per the minimum-station principle, both processes can be performed within one station assigned to C, increasing S_num to 3. Following this, [3,7]-[A,C] assigns processes 3 and 7 to workers A and C. Given that worker C already possesses a station (from processes 2/6), process 7 can be assigned to C's existing station. However, worker A requires a new station (distinct from it's first station for process 1), incrementing S_num to 4. This logic is applied iteratively to analyze the remaining assignments. The pseudo code for the HIP algorithm is provided in Table 4.

**Table 4. The pseudo code of HIP algorithm.**

**Algorithm 1 Heuristic Infeasibility Proof Algorithm**

01: **Input**: $\bar{x}_{wp}$, process_topo, $|S'|$, maxRS
02: **Output**: feasibility (True or False), min_p_set
03: **Initialize**: required_station_num←0, worker_set_before←∅, p_set←∅,
04:            min_p_set←set of all processes, feasibility←False, process_workers={p: [],...}
05: **Begin**
06: **for** $\forall$(w, p) ∈ $\bar{x}_{wp}$ **do**
07:    **if** $\bar{x}$[w,p]=1 **then**
08:       process_workers[p].append(w)
09:    **end if**
10: **end for**
11: **for** $\forall$pros ∈ process_topo **do**
12:    p_set←p_set∪set(pros)
13:    worker_set←get all workers of pros from process_workers
14:    **for** $\forall$w ∈ worker_set **do**
15:       **if** w ∉ worker_set_before and w **not** processed before current iteration **then**
16:          required_station_num←required_station_num+1
17:          **if** required_station_num> ($|S'|$ +maxRS) **and not** feasibility **then**
18:             min_p_set←p_set
19:             feasibility←True
20:          **end if**
21:       **end if**
22:    **end for**
23:    worker_set_before ← worker_set
24: **end for**
25: **return** feasibility, min_p_set
26: **End**

## 4.3 Enhanced combinatorial benders cut

G.Codato and M. Fischetti [29] break the original problem P into a master problem and a subproblem. If the master has an optimal solution $x^*$ and the subproblem has a solution $y^*$, then clearly $(x^*, y^*)$ is an optimal solution of the problem P. If the subproblem is infeasible, instead, $x^*$ itself is infeasible for problem P. So at least one binary variable has to be changed to break the infeasibility. This condition can be translated by the Inequality (31), namely the combinatorial Benders cut.

$$\sum_{i \in C: x^*_{j(i)}=0} x_j + \sum_{i \in C: x^*_{j(i)}=1} (1 - x_j) \geq 1 \tag{31}$$

Building upon the combinatorial Benders cut generation method introduced by G. Codato and M. Fischetti [29], we apply this approach to our model, resulting in Inequality (32). However, the large-scale nature of our problem, characterized by numerous worker-process combinations (specific dimensions are detailed in Table 8), presents a significant computational challenge. Generating cuts solely via Inequality (32) often necessitates an excessive number of iterations to converge to the optimal solution due to this scale.

$$\sum_{[(w,p)|x_{wp}=0, w \in W, p \in P]} x_{wp} + \sum_{[(w,p)|x_{wp}=1, w \in W, p \in P]} (1 - x_{wp}) \geq 1 \tag{32}$$

The preceding HIP algorithm demonstrates the utilization of local information infeasibility to determine the necessity of subproblem modeling and to steer the end-to-end solution direction. Critically, this "erroneous information" can be strategically leveraged beyond the subproblem to accelerate the coordinated solution process between the master and subproblem. Specifically, upon encountering subproblem infeasibility, we exploit this information to rapidly refine the master problem formulation. This leads us to modify the implementation of the cut generation from Inequality (32) to the enhanced Inequality (33). This no-good cut formulation provides a more precise identification of the root causes triggering subproblem infeasibility. The practical application of formulation (33) within our solution procedure is exemplified in line 23 of Table 5.

$$\sum_{[(w,p)|x_{wp}=0, w \in W, p \in P_{min}]} x_{wp} + \sum_{[(w,p)|x_{wp}=1, w \in W, p \in P_{min}]} (1 - x_{wp}) \geq 1 \tag{33}$$

As shown in Table 5, following the application of the HIP algorithm, if a solution is identified as feasible (i.e., without violating any constraints), we proceed to model and solve the corresponding subproblem. However, if an optimal solution cannot be obtained for the subproblem, we employ the Local Branching technique to further refine and optimize the model. Local Branching was first introduced by M. Fischetti and P. M. Pardalos [30], it is a heuristic algorithm designed for mixed-integer programming (MIP) problems. Its core concept revolves around restricting the solution search space to rapidly identify superior solutions within a local neighborhood. Traditional approaches employ "hard fixing" for variables (i.e., directly setting the values of variables), which may lead to a decline in the quality of solutions. In contrast, local branching adopts "soft fixing". It restricts the range of variable changes by adding slack constraints, allowing a certain degree of flexibility. While fixing the majority of variables, it retains a slack space for a small number of variables to strike a balance between search efficiency and solution quality.

$$\sum_{[(w,p)|x_{wp}=0, w \in W, p \in P_{min}]} x_{wp} + \sum_{[(w,p)|x_{wp}=1, w \in W, p \in P_{min}]} (1 - x_{wp}) \geq k \tag{34}$$

Specifically, Local Branching is an iterative procedure that initiates with a small integer parameter k. Through the incremental application of inequality (34), the method facilitates a gradual adjustment of the model, enabling an escape from

**Table 5. The pseudo code of solving procedure with enhanced benders cut.**

| Algorithm 2 Solving Procedure with Enhanced Benders Cut |
|---|
| 01: **Input**: SP_TL(subproblem time limit), TL(total time limit), process_topo, $|S'|$, maxRS |
| 02: **Output**: Assignment Solution |
| 03: **Initialize**: subproblem solution status(sub_status = False) |
| 04: **Begin** |
| 05: model = _Build Master Problem_() |
| 06: model, $\bar{x}_{wp}$ ← _Solve Master Problem_(model) |
| 07:   **while** not exceed TL and not sub_status optimal **do** |
| 08:      feasible_status, minimum_p_set = HIP($\bar{x}_{wp}$, process_topo, $|S'|$, maxRS) |
| 09:      **if** feasible_status is True **then** |
| 10:         sub_status = _Solve Subproblem_(SP_TL) |
| 11:        **if** sub_status is OPTIMAL **then** |
| 12:           return Assignment Solution # Global optimum reached |
| 13:        **else** |
| 14:           status = _local_branching(model, $\bar{x}_{wp}$)_ # Sub Problem (SP) is infeasible - > Local Branching |
| 15:          **if** status is OPTIMAL **then** |
| 16:          return Assignment Solution |
| 17:          **else** |
| 18:            model = _add_cb_cut($\bar{x}_{wp}$_, process_topo) # add CB cut with Inequality (32) |
| 19:            return _Solve Master Problem_(model) |
| 20:          **end if** |
| 21:        **end if** |
| 22:     **else** |
| 23:        model = _add_cb_cut($\bar{x}_{wp}$_, minimum_p_set) # add CB cut with Inequality (33) |
| 24:        return _Solve Master Problem_(model) |
| 25:     **end if** |
| 26:   **end while** |
| 27: **End** |

local search basins and an exploration of broader solution spaces to enhance solution quality. Building on the insights gained from the HIP algorithm regarding solution infeasibility, we leverage the *min_p_set* information to strategically guide these model refinements. This targeted approach consequently strengthens the effectiveness of the Local Branching algorithm. The detailed pseudo code for the Local Branching procedure is provided in Table 6.

## 5 Computational study

### 5.1 Problem specifications and solution methods

We present the computational evaluation of the proposed Enhanced Benders Decomposition (EBD) algorithm. All algorithms were implemented in Python and executed on a computer with an Intel Core i5-10300H CPU @ 2.50 GHz. We utilized 60 test instances derived from real-world industrial projects at Huawei, with the data source available in https://github.com/LiPanfei-Lab/CALBP-TS. Detailed scale parameters for each instance are provided in Table 7.

Due to the high dimensionality of the decision variables in the problem addressed by this paper, which renders solution by a single genetic or greedy algorithm impractical, we established three baseline algorithms for comparison: a monolithic **MILP** formulation, a hybrid approach combining Genetic Algorithm with Linear Programming (**GA+LP**), and a hybrid approach combining Greedy Algorithm with Linear Programming (**Greedy+LP**).

### 5.2 Parameter settings for the algorithms

For both the EBD and MILP algorithms, the models were solved using the commercial solver Gurobi. The optimization objective employs the sum of absolute differences in workload between all pairs of workers. To enhance interpretability of the objective value, we normalized it to the [0,1] range via Equations (35)-(38), which evaluate the maximum worker

**Table 6. The pseudo code of local branching algorithm.**

| Algorithm 3 Local Branching |
| --- |
| 01: **Input**: model, $\bar{x}_{wp}$, process_topo, $|S'|$, maxRS |
| 02: **Output**: Model objective value, Solution |
| 03: **Initialize**: sub_status = False(subproblem solution status), K_SET(local branching k parameters) |
| 04: **Begin** |
| 05: **for** $\forall k \in$ K_SET **do** |
| 06:    model ← add local branching cut by using $\bar{x}_{wp}$ # Inequality(34) |
| 07:    model, $x_{wp}$ ← SolveMasterProblem(model) # $x_{wp}$ is local assignment, different from $\bar{x}_{wp}$ |
| 08:    **if** $x_{wp} \neq \varnothing$ **then** |
| 09:       feasible, *min_p_set* ← HIP($x_{wp}$, process_topo, $|S'|$, maxRS) |
| 10:       **if** feasible **then** |
| 11:          status ← SolveSubproblem($x_{wp}$) |
| 12:          **if** status = OPTIMAL **then** |
| 13:             Clear branching cuts |
| 14:             **return** Model objective value, Solution # find an optimal solution |
| 15:          **else** |
| 16:             Clear branching cuts |
| 17:             *add_cb_cut($\bar{x}_{wp}$, process_topo)* # add CB cut with Inequality (32) |
| 18:       **else** |
| 19:          Clear branching cuts |
| 20:          *add_cb_cut($\bar{x}_{wp}$, process_topo)* # add CB cut with Inequality (33) |
| 21:    **else** |
| 22:       Clear branching cuts |
| 23:       **break** |
| 24: **end for** |
| 25: **return** None, None # if we can't find optimal solution, then we finally return None |
| 26: **End** |

workload relative to the average. As evident from Equation (38), a smaller value indicates improved solution quality. Parameters $w_1$ and $w_2$, representing the instance-specific *upph_weight* and *volatility_weight* respectively, are derived automatically from instance data, eliminating the need for manual configuration.

$$y_{max} = max_{w \in W} y_w \tag{35}$$

$$\bar{y}_w = \frac{1}{|W|} \sum_{w \in W} y_w \tag{36}$$

$$y_{std} = \sqrt{\frac{1}{|W|} \sum_{w \in W} (y_w - \bar{y}_w)^2} \tag{37}$$

$$eval = (1 - \frac{\bar{y}_w}{y_{max}}) \times w_1 + \frac{y_{std}}{\bar{y}_w} \times w_2 \tag{38}$$

To ensure a fair comparison between these baseline algorithms and our proposed EBD algorithm, all algorithms were subjected to a strict time limit of 180 seconds. Furthermore, recognizing that the monolithic MILP often fails to reach optimality within this 180-second window, we permitted it to return feasible solutions within a 10% optimality gap. Conversely, the maximum iteration count for the other two hybrid baselines was set to 1000, with detailed parameter configurations provided in Table 8.

**Table 7. Problem instances scale (|P|: process size, |S|: station size, |W|: worker size, MC: maximum cycle count).**

| Instance | Scale | Instance | Scale |
|---|---|---|---|
| 1 | \|P\|=83, \|S\|=20, \|W\|=14, MC=5 | 32 | \|P\|=61, \|S\|=24, \|W\|=15, MC=4 |
| 2\|3\|4 | \|P\|=45, \|S\|=20, \|W\|=8, MC=5 | 33 | \|P\|=84, \|S\|=32, \|W\|=19, MC=5 |
| 5 | \|P\|=68, \|S\|=20, \|W\|=11, MC=5 | 34 | \|P\|=58, \|S\|=32, \|W\|=23, MC=8 |
| 6 | \|P\|=63, \|S\|=20, \|W\|=11, MC=5 | 35\|36 | \|P\|=36, \|S\|=25, \|W\|=16, MC=4 |
| 7\|8 | \|P\|=28, \|S\|=24, \|W\|=14, MC=4 | 37 | \|P\|=28, \|S\|=26, \|W\|=15, MC=8 |
| 9 | \|P\|=28, \|S\|=26, \|W\|=14, MC=4 | 38 | \|P\|=74, \|S\|=23, \|W\|=12, MC=4 |
| 10\|14\|16 | \|P\|=45, \|S\|=23, \|W\|=15, MC=4 | 39 | \|P\|=68, \|S\|=23, \|W\|=12, MC=3 |
| 11 | \|P\|=39, \|S\|=27, \|W\|=17, MC=4 | 40\|41\|42 | P\|=45, \|S\|=20, \|W\|=9, MC=5 |
| 12 | \|P\|=39, \|S\|=24, \|W\|=17, MC=4 | 43 | \|P\|=80, \|S\|=20, \|W\|=10, MC=5 |
| 13 | \|P\|=41, \|S\|=25, \|W\|=18, MC=4 | 44 | \|P\|=72, \|S\|=23, \|W\|=12, MC=8 |
| 15 | \|P\|=45, \|S\|=23, \|W\|=14, MC=4 | 45 | \|P\|=78, \|S\|=23, \|W\|=12, MC=5 |
| 17 | \|P\|=67, \|S\|=32, \|W\|=19, MC=4 | 46 | \|P\|=76, \|S\|=23, \|W\|=12, MC=4 |
| 18 | \|P\|=26, \|S\|=26, \|W\|=15, MC=8 | 47 | \|P\|=71, \|S\|=23, \|W\|=12, MC=8 |
| 19 | \|P\|=63, \|S\|=53, \|W\|=25, MC=8 | 48\|49 | \|P\|=66, \|S\|=23, \|W\|=12, MC=5 |
| 20 | \|P\|=54, \|S\|=32, \|W\|=22, MC=8 | 50 | \|P\|=50, \|S\|=20, \|W\|=9, MC=5 |
| 21 | \|P\|=55, \|S\|=32, \|W\|=23, MC=8 | 51 | \|P\|=51, \|S\|=20, \|W\|=11, MC=8 |
| 22 | \|P\|=71, \|S\|=32, \|W\|=20, MC=8 | 52 | \|P\|=48, \|S\|=24, \|W\|=15, MC=8 |
| 23 | \|P\|=81, \|S\|=32, \|W\|=20, MC=8 | 53 | \|P\|=56, \|S\|=24, \|W\|=16, MC=5 |
| 24 | \|P\|=61, \|S\|=32, \|W\|=19, MC=8 | 54 | \|P\|=38, \|S\|=20, \|W\|=16, MC=4 |
| 25 | \|P\|=104, \|S\|=32, \|W\|=24, MC=8 | 55 | \|P\|=69, \|S\|=24, \|W\|=16, MC=8 |
| 26 | \|P\|=35, \|S\|=26, \|W\|=15, MC=8 | 56 | \|P\|=32, \|S\|=27, \|W\|=16, MC=8 |
| 27 | \|P\|=70, \|S\|=23, \|W\|=13, MC=8 | 57 | \|P\|=51, \|S\|=24, \|W\|=15, MC=8 |
| 28 | \|P\|=68, \|S\|=23, \|W\|=13, MC=5 | 58 | \|P\|=63, \|S\|=55, \|W\|=28, MC=8 |
| 29 | \|P\|=68, \|S\|=20, \|W\|=11, MC=6 | 59 | \|P\|=56, \|S\|=55, \|W\|=26, MC=8 |
| 30 | \|P\|=55, \|S\|=24, \|W\|=16, MC=8 | 60 | \|P\|=35, \|S\|=24, \|W\|=16, MC=4 |
| 31 | \|P\|=60, \|S\|=24, \|W\|=16, MC=4 | | |

**Table 8. Algorithm parameter settings.**

| Algorithm | Solver | Time Limit | Key Parameters | Solution | Explanations |
|---|---|---|---|---|---|
| EBD | Gurobi 10.0.1 | MP: *<br>SP: 120s<br>Total: 180s | Local branching K range(10, 90, 4)<br>Objective: the sum of absolute differences in workload between all pairs of workers | Optimal | MP: master problem<br>SP: subproblem |
| MILP | Gurobi 10.0.1 | Total: 180s | Objective: the sum of absolute differences in workload between all pairs of workers | Feasible (gap <= 10%) | Set "MIPFocus=1" and "MIPGap=0.1" to get feasible solutions as soon as possible. |
| GA+LP | Gurobi 10.0.1 (LP Part) | Total: 180s | pop_size: 2000<br>iterations: 1000<br>perc_elitism: 0.02<br>perc_mat: 0.5<br>mutation_rate: 0.2 | Feasible | 1. perc_elitism: percentage of the best individuals of the current generation that will carry over the next.<br>2. perc_mat: percentage of the best individuals of the current generation that will have a chance to be selected as a parent.<br>3. Use LP to solve the w-p assignments, and GA for the rest part. |
| Greedy+LP | Gurobi 10.0.1 (LP Part) | Total: 180s | iterations: 1000 | Feasible | Use LP to solve the w-p assignments, and Greedy based algorithm for the rest part. |

## 5.3 Computational results

### 5.3.1 Statistical analysis and visualization of runtime and solution quality.
The four algorithms were evaluated across 60 instances, with their runtime and optimization objective values detailed in Tables 9 and 10, respectively. Instances where no solution was found within the time limit are marked with a slash (/).

Based on the computation time of the EBD algorithm, we sorted the instances in ascending order and divided them into four groups according to quartile ranges: Small (Q1), Medium (Q2), Large (Q3), and Super-Large (Q4). Figs 13 and 14

**Table 9. Computational runtime of different algorithms on individual problem instances.**

| Instance | Runtime (s) | | | | Instance | Runtime (s) | | | |
|---|---|---|---|---|---|---|---|---|---|
| | GA+LP | Greedy+LP | MILP | EBD | | GA+LP | Greedy+LP | MILP | EBD |
| 1 | 30.1 | 12.7 | 25.8 | 9.1 | 32 | 82.8 | 56.3 | 180 | 47.3 |
| 2\|3\|4 | 4.7 | 2.4 | 1.9 | 1.4 | 33 | 32.3 | 17.6 | 180 | 9.5 |
| | 4.9 | 2.6 | 2.3 | 1.3 | | | | | |
| | 4.6 | 2.3 | 1.7 | 1.4 | | | | | |
| 5 | 23.6 | 15.7 | 7.5 | 6.1 | 34 | 21.5 | 14.4 | 180 | 9.0 |
| 6 | 8.1 | 3.8 | 5.6 | 2.3 | 35\|36 | 20.4 | 13.7 | 180 | 5.3 |
| | | | | | | 23.2 | 15.9 | 180 | 5.6 |
| 7\|8 | 88.3 | 62.2 | 77.8 | 51.9 | 37 | / | / | 180 | 66.3 |
| | 90.9 | 65.2 | 104.9 | 55.2 | | | | | |
| 9 | 109.2 | 68.1 | 79.3 | 54.3 | 38 | 9.4 | 4.1 | 6.6 | 2.2 |
| 10\|14\|16 | 4.5 | 2.9 | 180 | 1.9 | 39 | 4.6 | 2.5 | 3.8 | 1.7 |
| | 5.7 | 3.8 | 180 | 1.7 | | | | | |
| | 4.9 | 3.0 | 180 | 1.7 | | | | | |
| 11 | / | / | 180 | 177.2 | 40\|41\|42 | 6 | 3.3 | 2.7 | 1.5 |
| | | | | | | 5.6 | 3.1 | 2.5 | 1.4 |
| | | | | | | 5.5 | 2.8 | 2.5 | 1.6 |
| 12 | / | / | 180 | 165.9 | 43 | 6.3 | 4.5 | 4.1 | 3.2 |
| 13 | 5.4 | 4.5 | 180 | 1.9 | 44 | 18.8 | 8.2 | 51.1 | 5.6 |
| 15 | 5.6 | 3.9 | 41.8 | 1.6 | 45 | 9.9 | 4.9 | 6.8 | 3.1 |
| 17 | 13.5 | 6.1 | 180 | 4.4 | 46 | 9.2 | 3.7 | 5.6 | 2.9 |
| 18 | 109.7 | 75.9 | 38.6 | 14.8 | 47 | 15.6 | 9.3 | 12.9 | 6.8 |
| 19 | 96.3 | 44.4 | 180 | 32.6 | 48\|49 | 7.5 | 4.4 | 9.6 | 2.9 |
| | | | | | | 9.8 | 4.5 | 7.9 | 2.7 |
| 20 | / | / | 180 | 166.7 | 50 | / | / | / | / |
| 21 | 14.1 | 10.8 | 180 | 8.5 | 51 | 10.8 | 5.8 | 8.4 | 3.3 |
| 22 | 35.8 | 28.9 | 180 | 26.3 | 52 | 10.4 | 6.7 | 180 | 3.7 |
| 23 | 26.9 | 20.7 | 180 | 17.8 | 53 | 7.2 | 3.8 | 150.9 | 2.9 |
| 24 | 28.7 | 14.1 | 180 | 11.3 | 54 | / | / | / | / |
| 25 | / | / | / | / | 55 | 35.6 | 17.4 | 15.8 | 11.1 |
| 26 | 10.1 | 5.9 | 180 | 3.1 | 56 | / | / | / | 15.2 |
| 27 | 15.9 | 9.5 | 18.1 | 6.5 | 57 | 11.2 | 8.1 | 180 | 4.8 |
| 28 | 56.8 | 27.4 | 35.5 | 16.3 | 58 | / | / | / | / |
| 29 | 11.9 | 5.6 | 6.3 | 3.7 | 59 | / | / | / | 127.9 |
| 30 | 13.4 | 7.7 | 180 | 4.9 | 60 | 5.3 | 2.6 | 180 | 1.2 |
| 31 | 22.1 | 11.5 | 180 | 8.7 | | | | | |

depict runtime (excluding MILP due to consistent time-limit attainment) and solution quality across scaled test instances with complete algorithm data.

Using averaged data from instances within each quartile range, we calculated the Gap values between EBD and three benchmark algorithms in terms of both computation time and solution quality. The Gap is defined as: $Gap = (Baseline\_value - EBD\_value)/ Baseline\_value$. A higher Gap value indicates a greater performance improvement by EBD relative to the benchmark algorithms. Detailed results are presented in Tables 11 and 12.

**Table 10. Objective function values of different algorithms on individual problem instances.**

| Instance | Objective Value | | | | Instance | Objective Value | | | |
|---|---|---|---|---|---|---|---|---|---|
| | GA+LP | Greedy+LP | MILP | EBD | | GA+LP | Greedy+LP | MILP | EBD |
| 1 | 0.649 | 0.704 | 0.594 | 0.550 | 32 | 0.307 | 0.418 | 0.246 | 0.213 |
| 2\|3\|4 | 0.562 | 0.609 | 0.515 | 0.474 | 33 | 0.612 | 0.678 | 0.516 | 0.455 |
| | 0.567 | 0.596 | 0.498 | 0.485 | | | | | |
| | 0.559 | 0.611 | 0.517 | 0.474 | | | | | |
| 5 | 0.239 | 0.308 | 0.201 | 0.189 | 34 | 0.322 | 0.396 | 0.298 | 0.238 |
| 6 | 0.347 | 0.376 | 0.318 | 0.290 | 35\|36 | 0.305 | 0.419 | 0.238 | 0.208 |
| | | | | | | 0.262 | 0.283 | 0.238 | 0.208 |
| 7\|8 | 0.297 | 0.353 | 0.215 | 0.196 | 37 | 0.377 | 0.395 | 0.359 | 0.182 |
| | 0.263 | 0.344 | 0.215 | 0.196 | | | | | |
| 9 | 0.321 | 0.396 | 0.284 | 0.262 | 38 | 0.472 | 0.551 | 0.423 | 0.390 |
| 10\|14\|16 | 0.334 | 0.397 | 0.282 | 0.249 | 39 | 0.362 | 0.413 | 0.331 | 0.306 |
| | 0.347 | 0.415 | 0.282 | 0.249 | | | | | |
| | 0.307 | 0.332 | 0.282 | 0.249 | | | | | |
| 11 | 0.375 | 0.392 | 0.338 | 0.269 | 40\|41\|42 | 0.434 | 0.461 | 0.389 | 0.366 |
| | | | | | | 0.438 | 0.465 | 0.389 | 0.366 |
| | | | | | | 0.436 | 0.463 | 0.389 | 0.366 |
| 12 | 0.346 | 0.373 | 0.319 | 0.269 | 43 | 0.481 | 0.522 | 0.439 | 0.406 |
| 13 | 0.229 | 0.333 | 0.189 | 0.166 | 44 | 0.198 | 0.229 | 0.145 | 0.132 |
| 15 | 0.263 | 0.335 | 0.241 | 0.219 | 45 | 0.273 | 0.343 | 0.223 | 0.204 |
| 17 | 0.499 | 0.548 | 0.460 | 0.393 | 46 | 0.195 | 0.257 | 0.133 | 0.122 |
| 18 | 0.193 | 0.357 | 0.139 | 0.139 | 47 | 0.388 | 0.477 | 0.346 | 0.317 |
| 19 | 0.352 | 0.413 | 0.302 | 0.201 | 48\|49 | 0.414 | 0.492 | 0.362 | 0.347 |
| | | | | | | 0.417 | 0.442 | 0.362 | 0.347 |
| 20 | 0.252 | 0.321 | 0.224 | 0.181 | 50 | / | / | / | / |
| 21 | 0.317 | 0.367 | 0.277 | 0.202 | 51 | 0.332 | 0.439 | 0.283 | 0.267 |
| 22 | 0.245 | 0.319 | 0.191 | 0.142 | 52 | 0.297 | 0.388 | 0.236 | 0.212 |
| 23 | 0.243 | 0.316 | 0.225 | 0.175 | 53 | 0.272 | 0.295 | 0.249 | 0.226 |
| 24 | 0.271 | 0.319 | 0.223 | 0.179 | 54 | / | / | / | / |
| 25 | / | / | / | / | 55 | 0.312 | 0.454 | 0.241 | 0.219 |
| 26 | 0.275 | 0.295 | 0.248 | 0.205 | 56 | / | / | / | 0.178 |
| 27 | 0.211 | 0.286 | 0.166 | 0.152 | 57 | 0.221 | 0.238 | 0.204 | 0.171 |
| 28 | 0.391 | 0.438 | 0.327 | 0.304 | 58 | / | / | / | / |
| 29 | 0.255 | 0.315 | 0.215 | 0.199 | 59 | / | / | / | 0.184 |
| 30 | 0.268 | 0.329 | 0.221 | 0.187 | 60 | 0.441 | 0.473 | 0.407 | 0.332 |
| 31 | 0.514 | 0.555 | 0.473 | 0.411 | | | | | |

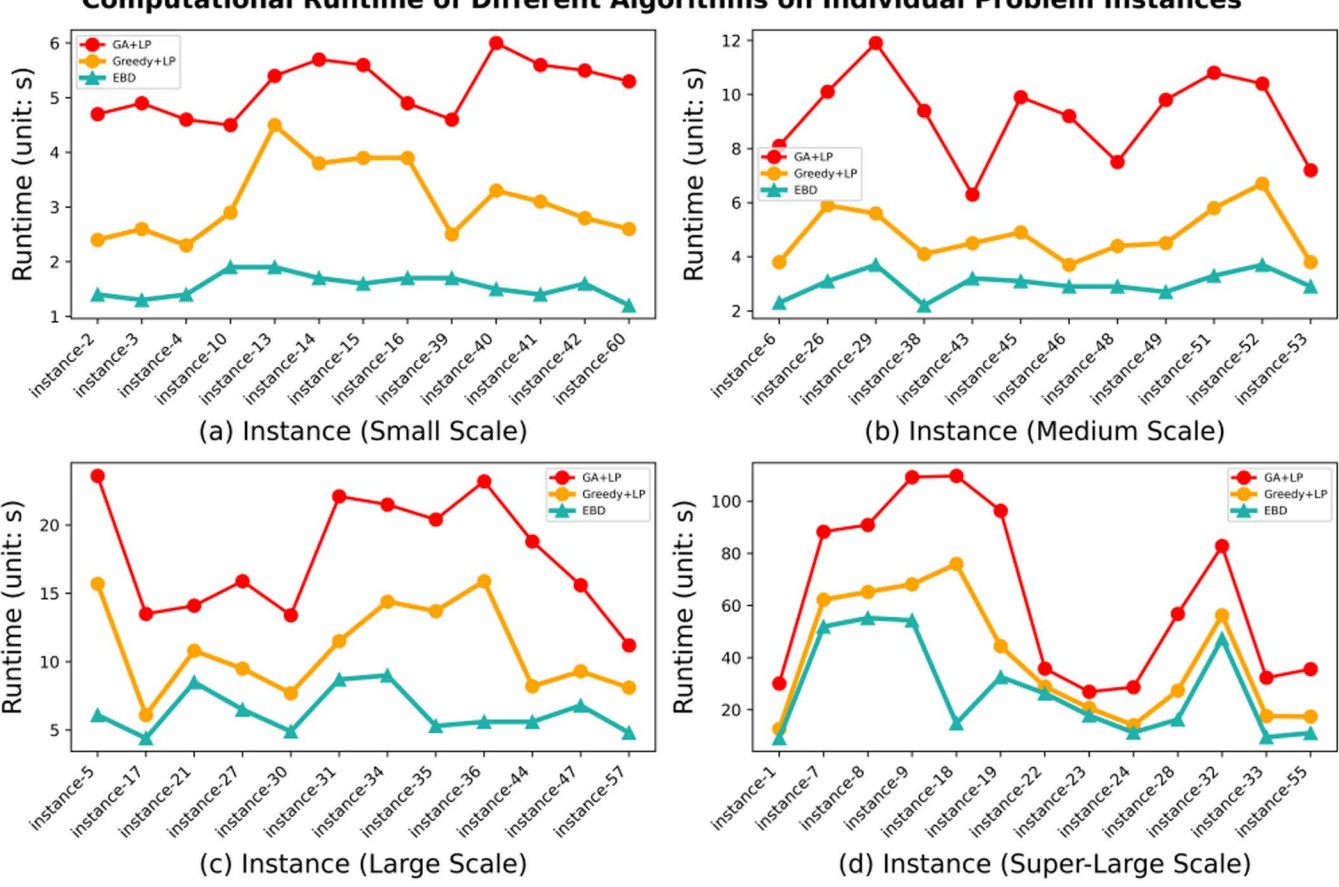

**Fig 13. Computational runtime of the algorithms across individual problem instances grouped by scale: (a) Small, (b) Medium, (c) Large, (d) Super-Large.**

**5.3.2 Ablation study.** To validate the proposed optimization strategy, we conducted ablation studies on instances where the original EBD algorithm requires iterative Benders cut generation between master and subproblems to obtain optimal solutions. The computational gap in runtime between the ablated strategies and the baseline EBD strategy per instance was quantified as: *Gap = (Ablated_value – EBD_value)/ EBD_value*, with negative values indicating efficiency degradation, as shown in Table 13.

### 5.4 Findings and discussion

As shown in Table 11, the EBD algorithm achieves significant reductions in computation time compared to Greedy+LP, GA+LP, and MILP. Specifically, the time reduction is most pronounced against MILP, followed by GA+LP, with the smallest improvement over Greedy+LP. Conversely, Table 12 demonstrates that solution quality improvement is greatest relative to Greedy+LP, followed by GA+LP, and smallest against MILP. This divergence stems from fundamental algorithmic characteristics: (i) The Greedy heuristic prioritizes rapid feasibility attainment at the expense of global optimality; (ii) The GA employs natural evolution principles, where solution quality depends on population size, iteration count, and problem complexity, outperforming the Greedy approach; (iii) Both MILP and EBD adopt global optimization strategies, but MILP exhibits the lowest computational efficiency for highly complex problems. Nevertheless, by enforcing a feasibility criterion of Gap ≤ 10%, solutions from MILP achieve the closest quality to those of EBD.

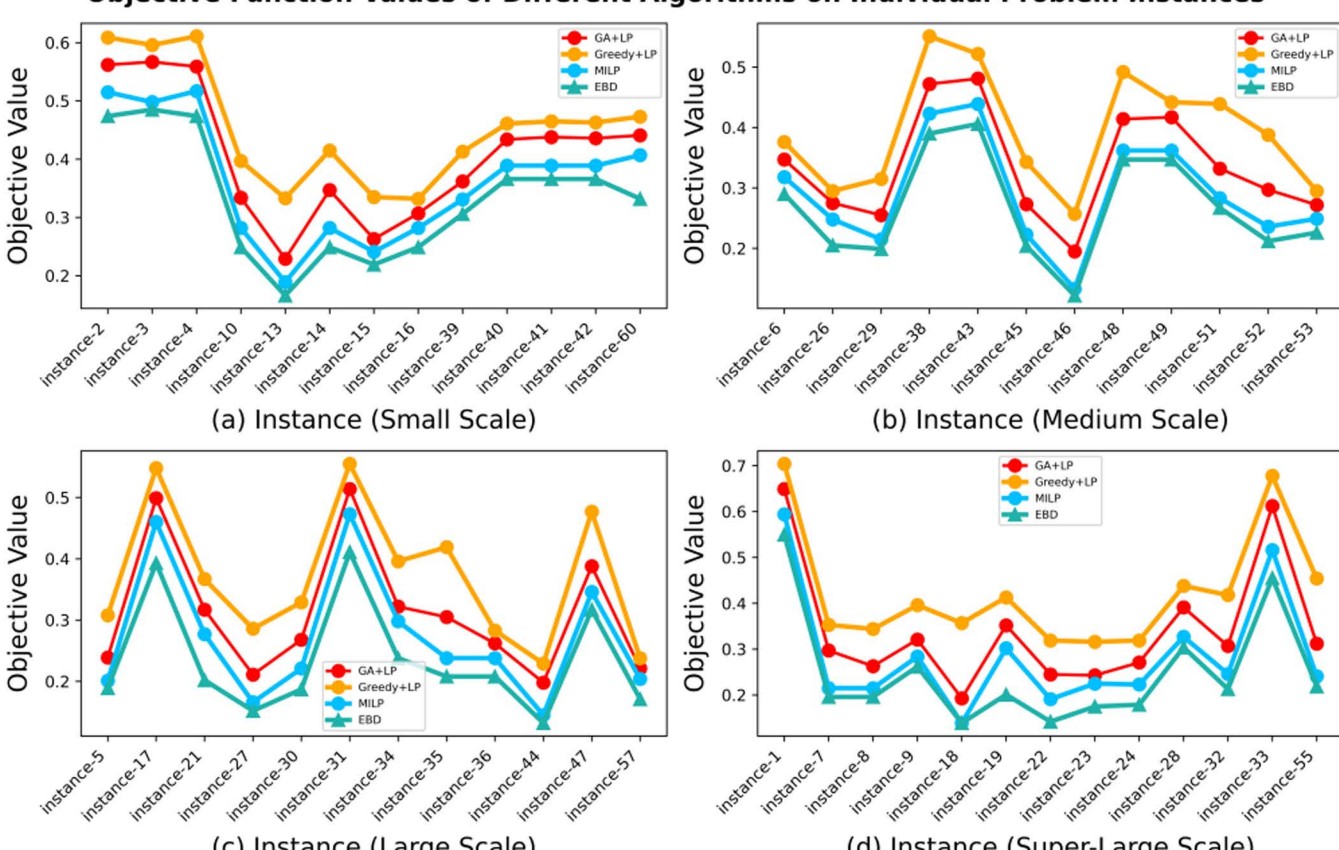

**Fig 14. Objective function values of the algorithms across individual problem instances grouped by scale: (a) Small, (b) Medium, (c) Large, (d) Super-Large.**

**Table 11. Average runtime (s) and gaps of algorithms on problem instances of varying scales.**

| Algorithm | Small Scale | | Medium Scale | | Large Scale | | Super-Large Scale | |
|---|---|---|---|---|---|---|---|---|
| | Avg. Runtime | Gap% | Avg. Runtime | Gap% | Avg. Runtime | Gap% | Avg. Runtime | Gap% |
| EBD | 1.6 | * | 3.0 | * | 6.4 | * | 27.5 | * |
| Greedy+LP | 3.1 | 48.4 | 4.8 | 37.5 | 10.9 | 41.3 | 39.3 | 30.0 |
| GA+LP | 5.2 | 69.2 | 9.2 | 67.4 | 17.8 | 64.0 | 63.3 | 56.6 |
| MILP | 73.8 | 97.8 | 47.7 | 93.8 | 127.5 | 94.9 | 112.1 | 75.5 |

Cross-scale analysis of benchmark instances reveals a divergence: the computational time advantage of EBD over benchmark algorithms diminishes with increasing instance scale, whereas its solution quality advantage intensifies. This bifurcation stems from algorithmic mechanisms. As problem scale grows, the iteration count between master and subproblems in EBD increases substantially, imposing heavier computational burden that erodes time-efficiency gains. Nevertheless, this iterative architecture outperforms traditional heuristics and MILP in solution refinement—since benchmark algorithms lack capability to diagnose infeasibility causes, they struggle to achieve comparable solution quality.

**Table 12. Average objectives and optimality gaps of algorithms on problem instances of varying scales.**

| Algorithm | Small Scale | | Medium Scale | | Large Scale | | Super-Large Scale | |
|---|---|---|---|---|---|---|---|---|
| | Obj. | Gap% | Obj. | Gap% | Obj. | Gap% | Obj. | Gap% |
| EBD | 0.331 | * | 0.268 | * | 0.234 | * | 0.249 | * |
| Greedy+LP | 0.454 | 27.1 | 0.393 | 31.8 | 0.369 | 36.6 | 0.424 | 41.3 |
| GA+LP | 0.406 | 18.5 | 0.336 | 20.2 | 0.312 | 25.0 | 0.343 | 27.4 |
| MILP | 0.363 | 8.8 | 0.291 | 7.9 | 0.272 | 13.9 | 0.286 | 12.9 |

As shown in Table 13, removing either the Heuristic Infeasibility Proof (HIP) or the Enhanced Benders Cut (EBC) generation strategy significantly degrades EBD's computational efficiency, increasing solution time. Simultaneous removal of both HIP and EBC further magnifies this detrimental effect. This verifies the synergistic effect of HIP and EBC in accelerating solution efficiency, achieving up to 19.1% reduction in runtime for certain instances.

# 6 Concluding remarks

The primary contribution of this study is the introduction of the Circular Assembly Line Balancing Problem with Task-Splitting (CALBP-TS), extending the research scope beyond traditional straight and U-shaped assembly line configuration.

From a theoretical perspective, we present a novel framework based on combinatorial Benders decomposition to address the complex four-dimensional (process-worker-station-machine) spatio-temporal assignment problem through decomposition into MP and SP. Considering the logical sequence of (worker-process) versus (process-station) decisions, two distinct decomposition strategies are designed: top-down (worker-process priority) and bottom-up (process-station priority). Experimental results demonstrate that the proposed framework achieves significant improvements in both computational time and solution quality compared to monolithic MILP, GA+LP, and Greedy+LP algorithms.

From a methodological perspective, to address the practical requirement of task splitting, an exact mathematical model quantifying workers' actual workloads in split-task scenarios is developed for the master problem, accompanied by rigorous theoretical proofs. For the subproblem, a dummy process encoding technique is introduced, which extends process representation and eliminates the need for explicit task-splitting modeling, thereby providing an end-to-end solution to this challenge.

**Table 13. Comparative runtime (s) analysis: ablation study of algorithmic techniques.**

| Instance | EBD | EBD – EBC | | EBD – HIP | | BD(EBD – EBC – HIP) | |
|---|---|---|---|---|---|---|---|
| | Runtime | Runtime | Gap% | Runtime | Gap% | Runtime | Gap% |
| 6 | 2.3 | 2.5 | −8.6 | 2.4 | −4.3 | 2.6 | −13.0 |
| 7 | 51.9 | 55.3 | −6.6 | 53.7 | −3.5 | 61.8 | −19.1 |
| 8 | 55.2 | 58.1 | −5.3 | 56.5 | −2.4 | 60.1 | −8.9 |
| 9 | 54.3 | 57.2 | −5.4 | 55.8 | −2.8 | 62.3 | −14.7 |
| 22 | 26.3 | 27.8 | −5.7 | 27.0 | −2.7 | 30.7 | −16.7 |
| 29 | 3.7 | 3.9 | −5.4 | 3.8 | −2.7 | 4.1 | −10.8 |
| 32 | 47.3 | 49.6 | −4.9 | 48.9 | −3.4 | 52.4 | −10.2 |
| 33 | 9.5 | 10.2 | −7.4 | 9.8 | −3.2 | 10.8 | −13.7 |
| 35 | 5.3 | 5.6 | −5.7 | 5.5 | −3.4 | 6.2 | −16.9 |
| 38 | 2.2 | 2.4 | −9.1 | 2.3 | −4.5 | 2.5 | −8.9 |
| 43 | 3.2 | 3.6 | −12.5 | 3.4 | −6.3 | 3.5 | −9.4 |
| 55 | 11.1 | 11.8 | −6.3 | 11.5 | −3.6 | 13.1 | −18.0 |
| Avg. | 22.7 | 24.0 | −5.7 | 23.4 | −3.1 | 25.8 | −13.7 |

After implementing the developed algorithms, several managerial insights are provided. The standard Benders decomposition is augmented with a Heuristic Infeasibility Proof (HIP) to effectively detect master problem solution feasibility and an Enhanced Benders Cut (EBC) generation algorithm that leverages information from infeasible solutions for efficient model refinement. Experimental results demonstrate significant improvements in both computational time and solution quality compared to MILP, GA+LP, and Greedy+LP algorithms. Ablation studies confirm that HIP and EBC individually enhance computational efficiency by 5.7% and 13.7%, respectively, on instances with strong MP-SP coupling, while their synergistic integration yields a further improvement, reaching 13.7%.

Despite these contributions, several limitations and potential avenues for future research remain.

- Adaptive Parameter Tuning: The iteration parameter K for Local Branching is currently fixed. Given the substantial variation in instance scale and complexity, future work should develop adaptive mechanisms to dynamically optimize K based on problem characteristics.

- Incorporating Additional Economic Objectives: While the current objective minimizes the sum of absolute differences in workload between all pairs of workers, future studies could incorporate additional economic objectives such as minimizing worker movement distance by leveraging the circular layout, optimizing scarce machine utilization, or reducing the number of stations required.

- Modeling Bundled Task Assignment: While the model supports task-level splitting, it currently lacks accommodation for bundled task assignment – a practical constraint where a worker assisting another must assume the entire set of the latter's processes. Incorporating this constraint into the framework represents a key direction for future research.

## Supporting information

**S1 File.** The raw statistical data underlying the key figures and tables.
(ZIP)

## Author contributions

**Conceptualization:** Panfei Li.

**Data curation:** Panfei Li.

**Formal analysis:** Panfei Li, Chongxing Ji.

**Funding acquisition:** Panfei Li.

**Investigation:** Panfei Li.

**Methodology:** Panfei Li.

**Project administration:** Panfei Li.

**Resources:** Panfei Li.

**Software:** Panfei Li.

**Supervision:** Panfei Li.

**Validation:** Panfei Li, Chongxing Ji.

**Visualization:** Panfei Li, Chongxing Ji.

**Writing – original draft:** Panfei Li.

**Writing – review & editing:** Panfei Li.

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
