## [Decision Letter · Decision Letter 0]

29 Jun 2025

PONE-D-25-23652Application of Enhanced Benders Decomposition Algorithm in Circular Assembly Line Balancing Problem with Task SplittingPLOS ONE

Dear Dr. li,

Thank you for submitting your manuscript to PLOS ONE. After careful consideration, we feel that it has merit but does not fully meet PLOS ONE’s publication criteria as it currently stands. Therefore, we invite you to submit a revised version of the manuscript that addresses the points raised during the review process.

 Manuscript needs to be revised according to the reviewer's comments. 

We look forward to receiving your revised manuscript.

Kind regards,

Jabir Mumtaz, Ph.D

Academic Editor

PLOS ONE

Journal Requirements:

“Industrial System Optimization, Operations Research, Deep Learning, and Intelligent Control.”

**Comments from PLOS Editorial Office** : We note that one or more reviewers has recommended that you cite specific previously published works in the current and previous rounds of revision. As always, we recommend that you evaluate the requested works to determine whether they are relevant and should be cited. It is not a requirement to cite these works and you may remove any added citations before the manuscript proceeds to publication. We appreciate your attention to this request.

**Additional Editor Comments:**

Need to be revised based on the reviewer's comments.

Reviewers' comments:

Reviewer's Responses to Questions

**Comments to the Author**

1. Is the manuscript technically sound, and do the data support the conclusions?

Reviewer #1: Yes

Reviewer #2: Yes

Reviewer #3: Yes

Reviewer #4: Partly

2. Has the statistical analysis been performed appropriately and rigorously? 

Reviewer #1: Yes

Reviewer #2: Yes

Reviewer #3: Yes

Reviewer #4: Yes

3. Have the authors made all data underlying the findings in their manuscript fully available?

Reviewer #1: Yes

Reviewer #2: Yes

Reviewer #3: Yes

Reviewer #4: Yes

4. Is the manuscript presented in an intelligible fashion and written in standard English?

Reviewer #1: Yes

Reviewer #2: No

Reviewer #3: Yes

Reviewer #4: No

5. Review Comments to the Author

Reviewer #1: This study has considerable potential to advance the academic literature from both theoretical and practical perspectives. I recommend acceptance once the following points are thoroughly addressed:

In the abstract, the authors state that the “EBD algorithm demonstrates superior computational efficiency and solution quality compared with traditional BD and GA approaches.” Please support this claim with brief numerical evidence.

The introduction should present up to three clear research questions so the paper’s motivation is easier to grasp.

Clarify the paper’s contributions under three aspects: (i) methodological, (ii) managerial, and (iii) theoretical.

The literature-review section needs a comprehensive table that (a) summarizes prior work, (b) highlights their gaps, and (c) shows how this study fills those gaps. The following highly relevant studies should also be reviewed:

An integrated bi-objective U-shaped assembly-line balancing and parts-feeding problem: optimization model and exact solution method

Robust optimization for U-shaped assembly-line worker assignment and balancing problem with uncertain task times

Assembly-line balancing by using axiomatic design principles: An application from the cooler-manufacturing industry

Provide a clear, straightforward numerical example in the problem-description section.

Discuss how the proposed Benders-decomposition-based algorithm could be applied to related problems (e.g., disassembly-line balancing, seru scheduling). Relevant references include:

Aggregated planning to solve a multi-product, multi-period disassembly-line balancing problem with multi-manned stations

Tactical-level strategies for a multi-objective disassembly-line balancing problem with multi-manned stations

Lot streaming in a workforce-scheduling problem for a seru production system under Shojinka philosophy

Seru scheduling with lot streaming and worker transfers: A multi-objective approach

In the computational analysis, consider a design-of-experiments framework to evaluate the impact of key controllable factors and their levels.

Add a sensitivity analysis to show how important factors and their levels influence the results.

Include a “Findings and Discussion” section that answers the research questions and interprets the results.

The conclusion should summarize contributions and findings, note limitations, and outline future research directions that address those limitations.

Consider renaming the conclusion to “Concluding Remarks.”

Reviewer #2: The authors developed an enhanced Benders Decomposition algorithm to solve a circular assembly line balancing problem. The details of the algorithm are presented. Numerical results validate the effectiveness of the algorithm. There are some minor comments should be considered before publication in the journal. (1) The manuscript should be proofread to make it clearer. (2) Some statements should be revised, For example, "ALBP resolution relied on manual scheduling protocols driven by human experts", "5 Computational Experience".

Reviewer #3: This paper proposes an enhanced Benders decomposition algorithm to solve the cyclic assembly line balancing problem with task splitting. The topic is interesting, but I believe a major revision is necessary before the paper can be considered for acceptance. My main concerns are as follows:

1. In the introduction, the authors directly emphasize that the cyclic assembly line balancing problem with task splitting is an important problem. However, it is evident that this is not a common type of assembly line. The authors should provide a more detailed explanation of the characteristics of such assembly lines, why this type of line is used, and what unique challenges it presents compared to conventional assembly lines.

2. Regarding the proposed enhanced Benders decomposition algorithm, the core contribution should be highlighted more clearly. Currently, the enhancement techniques appear to be quite standard, and it is difficult to identify the novel aspects of the proposed approach.

3. In the computational experiments, the authors should provide comparative analyses for each of the proposed enhancement strategies to validate their effectiveness. In addition, it would be valuable to compare the production efficiency and effectiveness of the proposed assembly line model with that of conventional assembly lines.

4. The conclusion section should be further enriched to better summarize the key findings and contributions of the paper.

Reviewer #4: The paper this paper proposes an Enhanced Benders Decomposition algorithm to solve the Circular Assembly Line Balancing Problem with Task Splitting. This topic is interesting. However, there are some drawbacks. Here are some comments for the authors.

1) The introduction section should be expanded and enriched, particularly in terms of introducing the research motivation.

2) The literature review section needs to be improved. The authors should include more relevant papers, such as An Improved Combinatorial Benders Decomposition Algorithm for the Human-Robot Collaborative Assembly Line Balancing Problem, and Combinatorial Benders decomposition for mixed-model two-sided assembly line balancing problem. A comparison table should be added.

3) The problem definition section requires rewriting due to certain confusing expressions. For instance, some parameters are listed in the notation table but are not utilized in the model. The authors mention parameters and variables related to machines, such as “IsMachineNeeded,” “Machines - indexed by m,” and “Maximum number of machines per station.” It is unclear whether any machines are actually used in CALBP-TS.

4) Compared to inequality (33), inequality (34) appears less strong; therefore, a comparative experiment is recommended.

5) The master problem is a multi-objective optimization problem. How can feasibility be checked and the objective value obtained (Figure 8 and 9)?

6) As shown in Figure 5, if the MP is infeasible, the algorithm terminates. This raises some confusion for me.

7) Why do you choose Pyomo and SCIP for solving the model instead of other mainstream solvers, such as Gurobi or CPLEX?

8) More experiments should be conducted to compare with other mainstream approaches in solving the problem.

6. PLOS authors have the option to publish the peer review history of their article (what does this mean? ). If published, this will include your full peer review and any attached files.

**Do you want your identity to be public for this peer review?** For information about this choice, including consent withdrawal, please see our Privacy Policy .

Reviewer #1: No

Reviewer #2: No

Reviewer #3: No

Reviewer #4: No

---

## [Author Response · Author response to Decision Letter 1]

9 Aug 2025

Dear Editor,

Thank you for all the reviewers’ suggestions, I will respond to these reviews step by step.

Reviewer #1: This study has considerable potential to advance the academic literature from both theoretical and practical perspectives. I recommend acceptance once the following points are thoroughly addressed:

We sincerely appreciate your constructive feedback. Below are our point-by-point responses

Q1: In the abstract, the authors state that the “EBD algorithm demonstrates superior computational efficiency and solution quality compared with traditional BD and GA approaches.” Please support this claim with brief numerical evidence.

We have added specific numerical evidence (Page 1, in the Abstract):

*"Validated on 60 real-world instances from Huawei, EBD achieves average runtime reductions of 97.8%, 69.2%, and 48.4% compared to MILP, GA+LP, and Greedy+LP baselines, respectively, while improving solution quality by up to 41.3%."*

Q2: The introduction should present up to three clear research questions so the paper’s motivation is easier to grasp.

The research questions (RQ1–RQ3) are now explicitly stated in Section 1 (Page 3, in the Introduction):

*"To address the CALBP-TS effectively, this study aims to answer the following key research questions:

• RQ1: How can the CALBP-TS be formally modeled to capture its unique characteristics (closed-loop topology, station revisitation, fixed-position machines, collaborative task execution) and high-dimensional complexity?

• RQ2: How can the combinatorial complexity arising from task-splitting (specifically, the non-linear workload allocation among multiple workers) be effectively resolved within an exact optimization framework, particularly in the master problem?

• RQ3: How can the computational efficiency of the classical Benders decomposition approach be significantly enhanced to solve large-scale CALBP-TS instances within practical time limits, especially through intelligent feasibility checks and cut generation?"*

Q3: Clarify the paper’s contributions under three aspects: (i) methodological, (ii) managerial, and (iii) theoretical.

Added in Section 1 as a bulleted list (Page 4, at the end of the Introduction):

*"To address the CALBP-TS effectively, this study aims to answer the following key research questions:

• From a theoretical standpoint, this paper introduces ......

• Regarding the methodology, we devise a mathematical model and integrated framework for CALBP-TS, ......

• From a managerial perspective, a Heuristic Infeasibility Proof (HIP) method to effectively detect infeasible solutions of the master problem is proposed. ......"*

Q4: The literature-review section needs a comprehensive table that (a) summarizes prior work, (b) highlights their gaps, and (c) shows how this study fills those gaps. The following highly relevant studies should also be reviewed:

Table 1 now includes:

Reviewed papers (e.g., Yılmaz 2020, Huang 2024).

Identified gaps (e.g., neglect of ergonomic factors, dynamic uncertainties).

This study’s contributions (e.g., circular topology, HIP-EBC synergy).

An integrated bi-objective U-shaped assembly-line balancing and parts-feeding problem: optimization model and exact solution method

Added:

Robust optimization for U-shaped assembly-line worker assignment and balancing problem with uncertain task times

Added:

Assembly-line balancing by using axiomatic design principles: An application from the cooler-manufacturing industry

As this paper exhibits low relevance to the core focus of our study and given that the existing references already provide sufficient coverage, we deemed its citation unnecessary. This decision was further supported by our inclusion of numerous other, more pertinent publications.

Q5: Provide a clear, straightforward numerical example in the problem-description section.

The following parts are added (Page 17-19, in “3.3 Numerical Example”):

In Section 3.3, we present numerical case studies to address two key challenges inherent to the application of our proposed combined Benders decomposition approach to CALBP-TS problems.

(1)Master Problem Modeling Challenge: The primary challenge within the master problem formulation lies in accurately calculating the workload of individual workers under task-splitting scenarios. To overcome this, we introduce a linearization modeling technique. Consequently, Section 3.3.1 provides the theoretical underpinnings for this formulation and demonstrates its application through a numerical example.

(2)Subproblem Simplification: Conversely, the subproblem formulation does not require explicit consideration of task splitting, owing to the implementation of our virtual operation encoding technique. Accordingly, Section 3.3.2 details a comprehensive numerical case study specific to this component.

Q6: Discuss how the proposed Benders-decomposition-based algorithm could be applied to related problems (e.g., disassembly-line balancing, seru scheduling). Relevant references include:

Aggregated planning to solve a multi-product, multi-period disassembly-line balancing problem with multi-manned stations

Tactical-level strategies for a multi-objective disassembly-line balancing problem with multi-manned stations

Lot streaming in a workforce-scheduling problem for a seru production system under Shojinka philosophy

Seru scheduling with lot streaming and worker transfers: A multi-objective approach

We sincerely appreciate your valuable suggestions. Upon in-depth examination, we found these cited works to be highly pertinent and well-executed. Notably, they address similar core challenges within the domain. In response to this insight, we have incorporated a discussion in the concluding section of the literature review (Section 2, Page 8), specifically exploring the potential for transferring the methodological framework proposed in this paper to address the related issues identified in these studies.

Q7: In the computational analysis, consider a design-of-experiments framework to evaluate the impact of key controllable factors and their levels.

Add a sensitivity analysis to show how important factors and their levels influence the results.

We sincerely appreciate your insightful suggestions regarding the experimental design. Upon careful reflection, we acknowledge that our initial approach had limitations. The core contribution of this paper lies in the application and innovation of an integrated framework for solving the CALBP-TS problem, specifically focusing on the efficient interaction between the master problem and subproblem.

Within this context, the key controllable factors evaluated in the Design of Experiments (DOE) framework are the individual and combined impacts of the Heuristic Infeasibility Proof (HIP) algorithm and the Enhanced Benders Cut (EBC) generation algorithm on the model's runtime.

Consequently, Section 5.3.2 (Page 32) has been expanded to include an ablation study. In this study, we define and compare four distinct algorithm configurations: EBD (baseline), EBD-EBC, EBD-HIP, and EBD-EBC-HIP, to rigorously assess their performance.

The experimental results indicate that the efficacy of these algorithmic components is primarily significant for a specific subset of instances characterized by strong master-subproblem coupling. Accordingly, Table 13 presents a detailed comparison of performance metrics for these particular instances, reporting both individual instance results and the average performance across this subset.

Q8: Include a “Findings and Discussion” section that answers the research questions and interprets the results.

This section has been thoroughly revised and enhanced to address the reviewer's comments and improve clarity, rigor, and alignment with the core contributions of the study. (Page 33):

Q9: The conclusion should summarize contributions and findings, note limitations, and outline future research directions that address those limitations.

This section has been thoroughly revised and enhanced to address the reviewer's comments and improve clarity, rigor, and alignment with the core contributions of the study. (Page 34)

Q10: Consider renaming the conclusion to “Concluding Remarks.”

The title has been revised in accordance with editorial advice to enhance clarity and better reflect the study's core focus and innovations.

Reviewer #2: The authors developed an enhanced Benders Decomposition algorithm to solve a circular assembly line balancing problem. The details of the algorithm are presented. Numerical results validate the effectiveness of the algorithm. There are some minor comments should be considered before publication in the journal.

Q: (1) The manuscript should be proofread to make it clearer. (2) Some statements should be revised, For example, "ALBP resolution relied on manual scheduling protocols driven by human experts", "5 Computational Experience".

We sincerely appreciate your valuable suggestions.

In response, the manuscript has undergone substantial revision and enhancement. Key improvements encompass linguistic precision, structural organization, heightened emphasis on core innovations, rigorous theoretical underpinnings of the models, in-depth critical analysis, comprehensive numerical case studies, comparative ablation studies for validation, and a significantly strengthened conclusion and future outlook. Given the extensive scope of these revisions, a detailed enumeration within this response is impractical. We therefore respectfully refer the reviewer to the comprehensively revised manuscript for a thorough examination of all enhancements.

Reviewer #3: This paper proposes an enhanced Benders decomposition algorithm to solve the cyclic assembly line balancing problem with task splitting. The topic is interesting, but I believe a major revision is necessary before the paper can be considered for acceptance. My main concerns are as follows:

We sincerely appreciate your insightful feedback, which will significantly enhance the academic rigor and impact of this work.

Q: 1. In the introduction, the authors directly emphasize that the cyclic assembly line balancing problem with task splitting is an important problem. However, it is evident that this is not a common type of assembly line. The authors should provide a more detailed explanation of the characteristics of such assembly lines, why this type of line is used, and what unique challenges it presents compared to conventional assembly lines.

The characteristics of such assembly lines (Introduction, Page 2):

*"The proposed Circular assembly lines (CALs) represent an advanced manufacturing configuration characterized by a closed-loop topology where stations are arranged in a circular layout, enabling processes to revisit stations across multiple production cycles."*

Why this type of line is used (Introduction, Page 2):

*"This structure is adopted in space-constrained industries (e.g., automotive, electronics) owing to its superior footprint efficiency and ability to accommodate fixed-position machines– immovable resources imposing constraints on conventional layouts."*

What unique challenges it presents compared to conventional assembly lines (Introduction, Page 2):

*"Unlike linear or U-shaped lines, Circular Assembly Lines (CALs) enable non-linear task sequencing by leveraging their closed-loop topology. This circumferencial movement allows workpieces to revisit stations or bypass others en route to fixed-machine locations, effectively decoupling task execution order from physical station sequence. This flexibility is further enhanced by proximity-based adjacency (including diametrically opposite stations), making CALs particularly suited for complex, hierarchical assemblies. However, these advantages introduce NP-hard challenges distinct from conventional ALBPs: 1) Revisitation overhead: Circumferential travel increases transport time, necessitating strict cycle limits (enforced by max_cycle_count); 2) Spatial dynamics: The circular layout complicates synchronization as workpieces dynamically traverse the loop, potentially skipping stations to reach required machines; 3) Task splitting (a core feature of CALBP-TS): Distributing bottleneck processes among multiple workers (up to max_worker_per_oper) improves balance but significantly amplifies combinatorial complexity, task synchronization overhead, and resource conflicts; 4) Worker mobility constraints: Assigning non-adjacent stations (limited by max_station_per_worker) incurs movement penalties not present in unidirectional systems. Collectively, the inherent cyclicity, dynamic routing, task splitting, and worker mobility constraints exponentially increase solution-space dimensionality, demanding novel optimization approaches."*

Q: 2. Regarding the proposed enhanced Benders decomposition algorithm, the core contribution should be highlighted more clearly. Currently, the enhancement techniques appear to be quite standard, and it is difficult to identify the novel aspects of the proposed approach.

In the Introduction, we systematically articulate the innovations of this work. (Introduction, Page 4)

The Experiments section then provides comprehensive validation through rigorous comparative analyses and in-depth discussions, demonstrating both the effectiveness and efficiency of the proposed methodology. (Computational Study, Page 32)

Finally, the Conclusion consolidates the core advancements by contextualizing their significance within the broader research landscape. (Concluding Remarks, Page 34)

Q: 3. In the computational experiments, the authors should provide comparative analyses for each of the proposed enhancement strategies to validate their effectiveness. In addition, it would be valuable to compare the production efficiency and effectiveness of the proposed assembly line model with that of conventional assembly lines.

We sincerely appreciate your insightful suggestions regarding the experimental design. Upon careful reflection, we acknowledge that our initial approach had limitations. The core contribution of this paper lies in the application and innovation of an integrated framework for solving the CALBP-TS problem, specifically focusing on the efficient interaction between the master problem and subproblem.

Within this context, the key controllable factors evaluated in the Design of Experiments (DOE) framework are the individual and combined impacts of the Heuristic Infeasibility Proof (HIP) algorithm and the Enhanced Benders Cut (EBC) generation algorithm on the model's runtime.

Consequently, Section 5.3.2 (Page 32) has been expanded to include an ablation study. In this study, we define and compare four distinct algorithm configurations: EBD (baseline), EBD-EBC, EBD-HIP, and EBD-EBC-HIP, to rigorously assess their performance.

The experimental results indicate that the efficacy of these algorithmic components is primarily significant for a specific subset of instances characterized by strong master-subproblem coupling. Accordingly, Table 13 presents a detailed comparison of performance metrics for these particular instances, reporting both individual instance results and the average performance across this subset.

Q: 4. The conclusion section should be further enriched to better summarize the key findings and contributions of the paper.

This section has been thoroughly revised and enhanced to address the reviewer's comments and improve clarity, rigor, and alignment with the core contributions of the study. (Page 34)

Reviewer #4: The paper this paper proposes an Enhanced Benders Decomposition algorithm to solve the Circular Assembly Line Balancing Problem with Task Splitting. This topic is interesting. However, there are some drawbacks. Here are some comments for the authors.

We sincerely appreciate your insightful feedback, which will significantly enhance the academic rigor and impact of this work.

Q: 1) The introduction section should be expanded and enriched, particularly in terms of introducing the research motivation.

We sincerely appreciate your constructive feedback. In response, we have substantially revised the Introduction to: (1) systematically establish the operational characteristics of Circular Assemb

---

## [Decision Letter · Decision Letter 1]

12 Sep 2025

Application of Enhanced Benders Decomposition Algorithm in Circular Assembly Line Balancing Problem with Task Splitting

PONE-D-25-23652R1

Dear Dr. li,

We’re pleased to inform you that your manuscript has been judged scientifically suitable for publication and will be formally accepted for publication once it meets all outstanding technical requirements.

Kind regards,

Jabir Mumtaz, Ph.D

Academic Editor

PLOS ONE

Additional Editor Comments (optional):

Reviewer #1:

Reviewer #2:

Reviewer #3:

Reviewer #4:

Reviewers' comments:

Reviewer's Responses to Questions

**Comments to the Author**

1. If the authors have adequately addressed your comments raised in a previous round of review and you feel that this manuscript is now acceptable for publication, you may indicate that here to bypass the “Comments to the Author” section, enter your conflict of interest statement in the “Confidential to Editor” section, and submit your "Accept" recommendation.

Reviewer #1: All comments have been addressed

Reviewer #2: All comments have been addressed

Reviewer #3: All comments have been addressed

Reviewer #4: All comments have been addressed

2. Is the manuscript technically sound, and do the data support the conclusions?

Reviewer #1: Yes

Reviewer #2: Yes

Reviewer #3: Yes

Reviewer #4: Yes

3. Has the statistical analysis been performed appropriately and rigorously? 

Reviewer #1: Yes

Reviewer #2: Yes

Reviewer #3: Yes

Reviewer #4: Yes

4. Have the authors made all data underlying the findings in their manuscript fully available?

Reviewer #1: Yes

Reviewer #2: Yes

Reviewer #3: Yes

Reviewer #4: Yes

5. Is the manuscript presented in an intelligible fashion and written in standard English?

Reviewer #1: Yes

Reviewer #2: Yes

Reviewer #3: Yes

Reviewer #4: Yes

6. Review Comments to the Author

Reviewer #1: The authors have considered all comments and conducted a substantial revision

Therefore I suggest to accept this manuscript for publication

Reviewer #2: The comments have been well responded and addressed. Now I think that It can be accepted for publication.

Reviewer #3: (No Response)

Reviewer #4: The authors have thoroughly addressed all my questions, and I believe the paper is ready for acceptance.

7. PLOS authors have the option to publish the peer review history of their article (what does this mean? ). If published, this will include your full peer review and any attached files.

**Do you want your identity to be public for this peer review?** For information about this choice, including consent withdrawal, please see our Privacy Policy .

Reviewer #1: No

Reviewer #2: No

Reviewer #3: No

Reviewer #4: No

---

## [Editor Report · Acceptance letter]

PONE-D-25-23652R1

PLOS ONE

Dear Dr. li,

I'm pleased to inform you that your manuscript has been deemed suitable for publication in PLOS ONE. Congratulations! Your manuscript is now being handed over to our production team.

Kind regards,

on behalf of

Dr Jabir Mumtaz

Academic Editor

PLOS ONE